# Single-cell sequencing maps gene expression to mutational phylogenies in PDGF- and EGF-driven gliomas

Sören Müller[1,2,†], Siyuan John Liu[1,2,†], Elizabeth Di Lullo[2,3], Martina Malatesta[1,2], Alex A Pollen[2,3], Tomasz J Nowakowski[2,3], Gary Kohanbash[1], Manish Aghi[1], Arnold R Kriegstein[2,3], Daniel A Lim[1,2,4,*] & Aaron Diaz[1,2,**]

## Abstract

Glioblastoma multiforme (GBM) is the most common and aggressive type of primary brain tumor. Epidermal growth factor (EGF) and platelet-derived growth factor (PDGF) receptors are frequently amplified and/or possess gain-of-function mutations in GBM. However, clinical trials of tyrosine-kinase inhibitors have shown disappointing efficacy, in part due to intra-tumor heterogeneity. To assess the effect of clonal heterogeneity on gene expression, we derived an approach to map single-cell expression profiles to sequentially acquired mutations identified from exome sequencing. Using 288 single cells, we constructed high-resolution phylogenies of EGF-driven and PDGF-driven GBMs, modeling transcriptional kinetics during tumor evolution. Descending the phylogenetic tree of a PDGF-driven tumor corresponded to a progressive induction of an oligodendrocyte progenitor-like cell type, expressing pro-angiogenic factors. In contrast, phylogenetic analysis of an *EGFR*-amplified tumor showed an up-regulation of pro-invasive genes. An in-frame deletion in a specific dimerization domain of PDGF receptor correlates with an up-regulation of growth pathways in a proneural GBM and enhances proliferation when ectopically expressed in glioma cell lines. In-frame deletions in this domain are frequent in public GBM data.

**Keywords** copy-number variation; glioblastoma; PDGFRA; single-cell RNA-sequencing; tumor phylogeny
**Subject Categories** Cancer; Chromatin, Epigenetics, Genomics & Functional Genomics; Genome-Scale & Integrative Biology
**Mol Syst Biol. (2016) 12: 889**

## Introduction

Glioblastoma multiforme (GBM) is an extremely aggressive type of brain tumor, characterized by a high degree of intra-tumor heterogeneity (Patel *et al*, 2014). Amplifications and gain-of-function mutations in receptor-tyrosine kinases (RTKs) are common in GBM. However, these mutations are typically regional and mosaic (Szerlip *et al*, 2012), and combinatorial application of RTK inhibitors is required to achieve a complete treatment *in vitro* (Stommel *et al*, 2007). Clinical trials of RTK inhibitors have shown only minimal efficacy, and these limitations may be in part due to intra-tumor heterogeneity (Prados *et al*, 2015). More broadly, developing treatments that circumvent specific, regional genotypic differences is challenging, since the number of biopsies per tumor is generally limited, and bulk-sequencing methods reduce such regional variation to population averages.

There is strong evidence that treatment itself can drive clonal evolution. Temozolomide chemotherapy is part of the current standard of care for newly diagnosed GBM. But, for some glioma patients, temozolomide treatment can also drive a hyper-mutated phenotype, as has been demonstrated by phylogenetic analysis of exome-sequencing (exome-seq) data (Johnson *et al*, 2014). Phylogenetic analyses of bulk exome-seq and methylation array data in glioma cohorts have also been used to identify recurrent events in tumor evolution (Johnson *et al*, 2014; Mazor *et al*, 2015). Recent advances in single-cell sequencing have enabled fine mapping of *EGFR* variant heterogeneity (Francis *et al*, 2014), as well as studies of the evolutionary history of individual tumors at unprecedented resolution (Navin *et al*, 2011; Garvin *et al*, 2014). Furthermore, large-scale copy-number variations (CNVs) have been inferred from single-cell RNA sequencing (RNA-seq) in GBM (Patel *et al*, 2014). In this study, we called CNVs from bulk exome-seq and quantified them in single-cell

1 Department of Neurological Surgery, University of California San Francisco, San Francisco, CA, USA
2 Eli and Edythe Broad Center of Regeneration Medicine and Stem Cell Research, University of California San Francisco, San Francisco, CA, USA
3 Department of Neurology, University of California San Francisco, San Francisco, CA, USA
4 Veterans Affairs Medical Center, San Francisco, CA, USA
  *Corresponding author. Tel: +1 415 502 2885; E-mail: daniel.lim@ucsf.edu
  **Corresponding author. Tel: +1 415 514 0408; E-mail: aaron.diaz@ucsf.edu
  †These authors contributed equally to this work

RNA-seq from the same tumor sample. Based on this, we produced a clonal ordering of individual cells that we used to infer transcriptional kinetics during tumor evolution and to perform inter-clone differential transcriptomics. We used this approach to contrast EGF-driven and PDGF-driven GBMs and identified pathways that show a dose–response to EGF- and PDGF-receptor copy number.

# Results

### Primary GBMs contain heterogeneous mixtures of cell types with recurrent transcriptional signatures

We collected fresh tissue from three cases of primary, untreated GBM directly from the operating room (SF10282, SF10345, and SF10360) and subjected these biopsies to both single-cell RNA-seq and bulk exome-seq (Fig 1A). We also performed bulk exome-seq on a separate blood sample from each patient. We characterized the landscape of genomic, somatic mutations for each patient using a robust exome-seq pipeline (Johnson *et al*, 2014), this analysis included identifying single nucleotide variants (SNVs), small insertions/deletions (indels), and copy-number variants (CNVs; Materials and Methods).

All cases demonstrated an amplification of growth factor genes. We found *EGFR* to be highly amplified in SF10345 (122 copies) and *PDGFRA* to be amplified in SF10282 (12 copies). Deletion and putative loss-of-function mutations in tumor suppressors were also common events (Datasets EV1–EV6). For example, all cases had non-synonymous point mutations in *PTEN* (with variant allele frequencies (VAFs) from 41 to 89%). A copy of chromosome 10 was lost in SF10345 and SF10360. Furthermore, these two cases harbored a deletion in chromosome 9, in a region encoding tumor suppressor genes *KLHL9* and *CDKN2A/B*. *KLHL9* deletions are correlated with the mesenchymal GBM subtype and poor prognosis (Chen *et al*, 2014). In our data, *KLHL9* is not expressed in either SF10345 or SF10360, and both samples classify as mesenchymal/classical. SF10360 and SF10282 share other mutations, such as a loss of 13q14 that contains the tumor suppressive micro-RNA cluster miR-15a/16 (Aqeilan *et al*, 2010; Afonso-Grunz & Müller, 2015). Between 5 and 35, small indels were detected per sample, for example, *TP53* (SF10282, frame-shift deletion), *NF1* (SF10360, frame-shift deletion), and *PLAGL1* (SF10345, frame-shift deletion).

Prior to the analysis of single-cell RNA-seq libraries, low-complexity and low-coverage libraries were filtered (Fig EV1A and B), and stromal/non-malignant cells were identified (Materials and Methods). This workflow left 61, 66, and 63 tumor cells from SF10282, SF10345, and SF10360, respectively. Consistent with previous reports (Patel *et al*, 2014), classification of single cells according to the Verhaak subtypes (Verhaak *et al*, 2010) identified heterogeneous mixtures of distinct subtypes within the same tumor (Fig 1C). SF10345 and SF10360 are classical/mesenchymal and predominantly EGFR driven. SF10282 is predominantly pro-neural, up-regulates PDGF-pathway genes, and markers of oligodendrocyte progenitor cells (OPCs) are broadly expressed. Yet SF10282 contains a subpopulation of cells with a neural stem cell (NSC)-like expression profile (Fig 1C). These cells classify as

mesenchymal/classical in the Verhaak scheme. We sought to infer the relative ordering of the NSC and OPC-like cells in the tumor's phylogeny and more generally to establish cellular phylogenies for all samples.

### Phylogenies of copy-number alterations map gene expression to clonal structure

We chose to focus on large, somatic CNVs of 100 exons or more (Materials and Methods). The median size of CNVs exceeding this threshold was 18–21 mega base-pairs, comprising 300–400 genes. This size is much larger than the size of CNVs previously observed to occur frequently in the germline (Sudmant *et al*, 2015), which had a median size of 36 kilo base-pairs. We found that GBM to normal-brain control single-cell expression ratios correlated with CNV status (Appendix Fig S1), motivating us to quantify these CNVs in individual cells (Materials and Methods). Briefly, for each CNV identified in the exome-seq, the 5% significance level of the distribution of normal-brain read counts covering that locus was used as a threshold to assign CNV presence/absence calls to individual cells (Fig 2A). This triage was unaffected by the application of a wavelet-smoothing filter to the single-cell data. Furthermore, exome-seq read histograms showed excellent agreement with single-cell trend-lines (Fig 2B and C). This indicated that our approach was robust to the stochastic expression of individual genes. We validated the error rate of this classifier using 10-fold cross-validation, as well as empirical testing on a control dataset (Pollen *et al*, 2015; Appendix Fig S2, Materials and Methods). Using Jaccard distance between CNV genotypes to assess inter-cell similarity, we fit phylogenies to each of the tumor samples using the Fitch–Margoliash method (Materials and Methods). The amplifications of chromosomes 7 and 19p13.3, which were shared across cases in the exome-seq, occurred early in all three of our single-cell phylogenies. In SF10360, chromosome 7 gain was a founding event, occurring together with a loss of chromosome 10 (Fig 3A). Intriguingly, a loss of chromosome 13 arose independently in two distinct sub-clones of SF10360. Since 13q14 harbors the miR-15a/16 micro-RNA cluster, a known tumor suppressor in prostate cancer (Bonci *et al*, 2008) and chronic lymphocytic leukemia (Pekarsky & Croce, 2015), this loss may convey a survival advantage here as well.

### MiR-15a/16 mutant cells up-regulate downstream adhesion and Aurora B kinase pathway genes, enriched in the leading edge and infiltrating tumor

We compared gene expression in chromosome 13 wild-type cells to cells harboring the deletion in SF10360 (Dataset EV7) and scanned the promoters of differentially expressed genes ($P < 0.05$) for conserved transcription factor recognition motifs (Materials and Methods). 16% of genes up-regulated upon chromosome 13 deletion were validated direct targets of miR-15a/16; and an additional 78% were expressed from loci enriched for motifs of transcription factors targeted by miR-15a/16 (Fig 3B; Chou *et al*, 2015). The most significant differentially expressed genes (adjusted $P < 0.1$) showed distinct patterning across tumor anatomical structures, when cross-referenced with the Ivy Glioblastoma Atlas

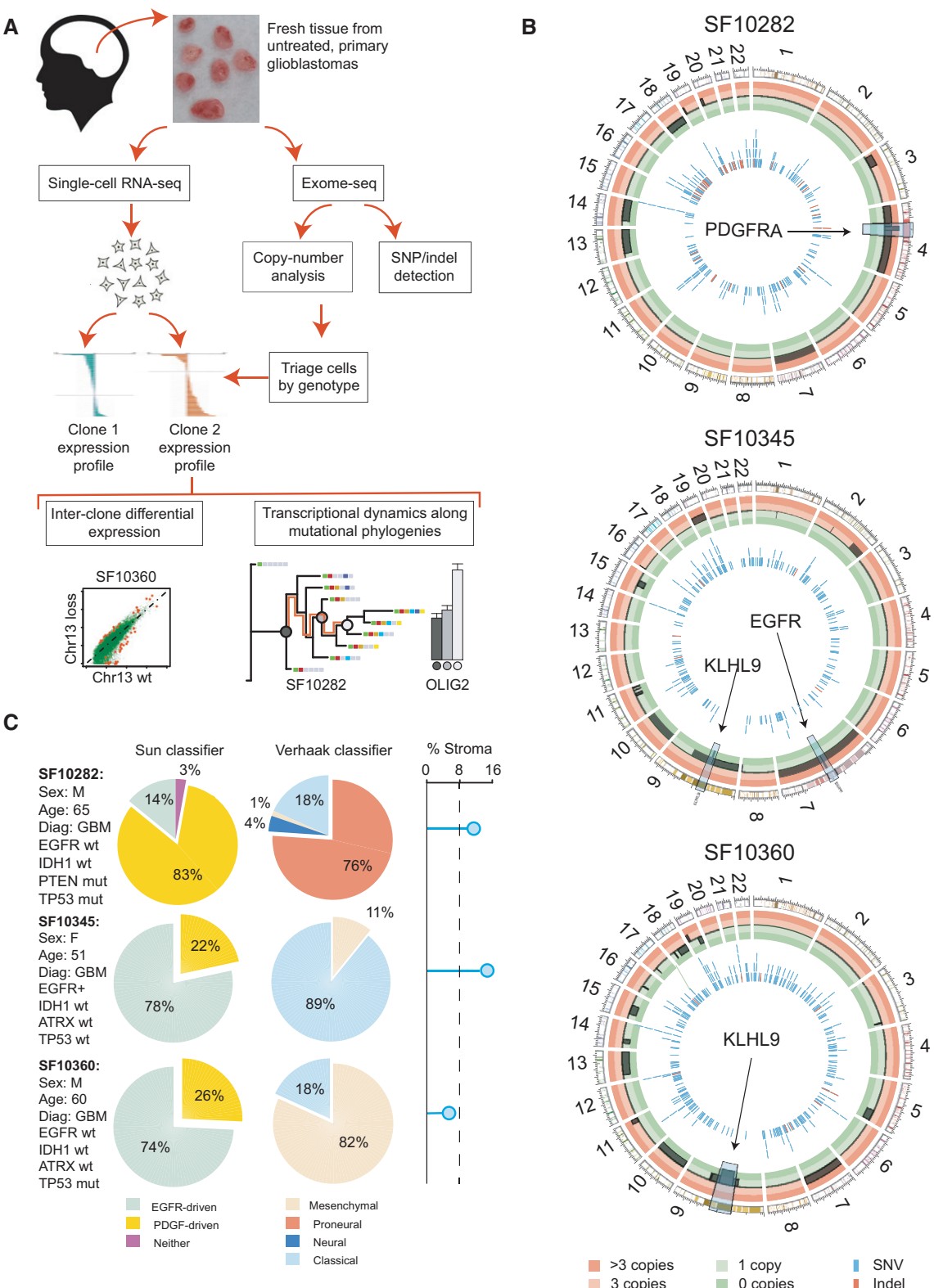

**Figure 1.  Experimental design and pipeline, pathology, and genomic and transcriptomic signatures for three primary GBMs.**

A   The sample acquisition and processing pipeline.

B   Circos plot of somatic, genomic alterations detected in the bulk DNA of each patient using ADTEx. Copy-number alterations are highlighted in the outer circle by thick black bars, and SNVs (MuTect) and small indels (Pindel) by the vertical lines in the inner circle. Regions with strong amplifications/losses are highlighted.

C   Summaries of the patient's sex, age, pathology report, sample's stromal infiltration, and molecular classification.

# A

## Single-cell CNV triage
## chr 19 : 2.81e+06 - 8.92e+6

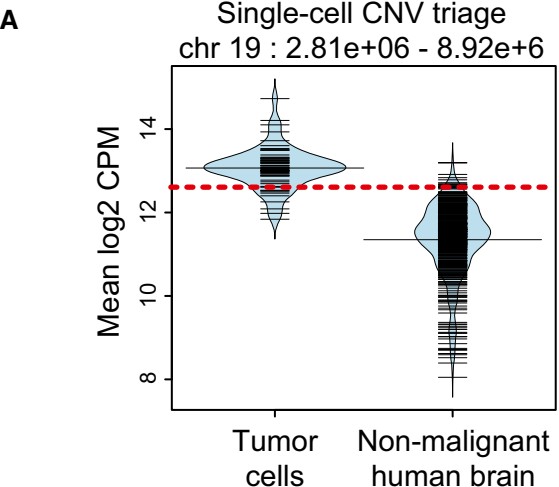

# B

## Exome-seq CNV calls

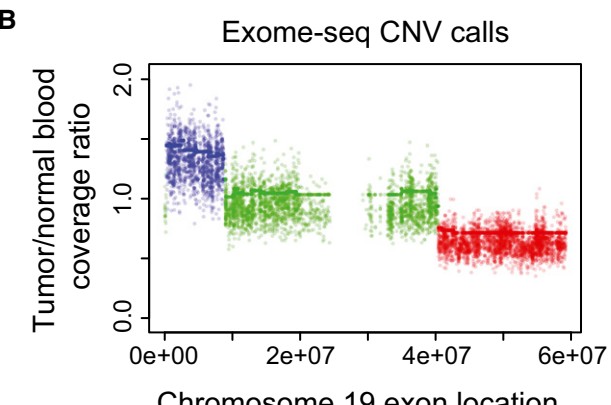

# C

## Single-cell CNV

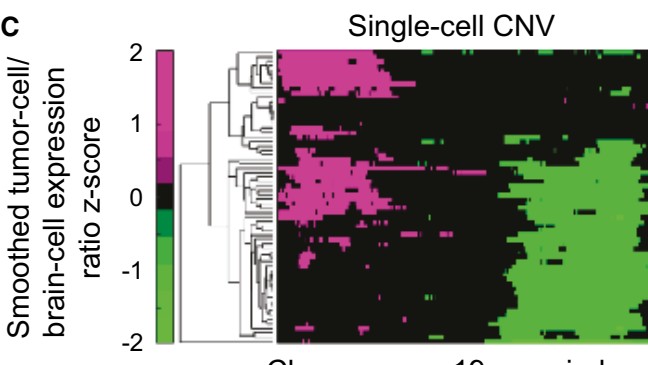

---

**Figure 2.  Presence/absence assignments in individual cells of CNVs called from bulk exome-seq.**

A  Read-count distributions in a locus of copy-number gain on 19p13.3, comparing cells in SF10360 to a human, normal-brain control. The 5th percentile of the normal-cell distribution is indicated by a red line.

B  Per-exon, normalized bulk exome-seq read-count computed by ADTEx, compared between SF10360 and a blood control. Regions with a read-count ratio > 1.3 are putative DNA copy-number gains (blue), and regions with fold-changes < 0.7 are putative losses (red).

C  A heatmap visualizing the z-score of ratios, of wavelet-smoothed read counts, compared between single-cell RNA-seq expression in tumor cell (rows) and the median expression in single-cell RNA-seq of normal-brain tissue.

---

(glioblastoma.alleninstitute.org). Genes up-regulated in the wild-type cells were enriched in the peri-vascular region, and genes up-regulated upon chromosome 13 deletion were enriched in the leading edge and infiltrating tumor (Figs 3C and EV2). Consistent with an infiltrating phenotype, cell-adhesion molecules were over-represented ($q = 0.06$; Materials and Methods). This included junction-adhesion molecules, integrins, disintegrins, and cell-surface receptors implicated in invasion (Fig 3D; Nath *et al*, 2000; Sloan, 2005; Tenan *et al*, 2010; Reyes *et al*, 2013; Sarkar *et al*, 2015; Venkatesh *et al*, 2015). 76% of these cell-adhesion pathway genes were enriched for NF-κB/REL recognition motifs (Fig 3B). Taken together with the EGFR-driven/mesenchymal classification of the case, we speculated that the loss of the miR-15a/16 cluster enhances growth factor-stimulated cell invasion here, as has been described in other cancers (Bonci *et al*, 2008). Among the direct targets of miR-15a/16 that were up-regulated, Aurora B kinase, survivin, and genes that complex with them were overrepresented ($q = 0.03$; Fig 3E). Survivin is a well-studied inhibitor of apoptosis. Aurora B kinase overexpression increases genomic instability (Ota *et al*, 2002), resulting in multinuclearity and aneuploidy (Tatsuka *et al*, 1998).

## Dose–response analysis correlates an in-frame deletion in *PDGFRA* to a pro-growth signature *in vivo*

We found that the PDGF-receptor alpha encoding gene, *PDGFRA*, was amplified in SF10282 (12 copies as estimated by exome-seq). We also detected a small deletion in exon 7 that was broadly expressed (Fig 4A). This mutant transcript, which we denote as *PDGFRA*$\Delta^7$, is found in 98% of SF10282's cells that express *PDGFRA* (69% of cells overall). Since the deletion is in-frame, broadly expressed, and affects an immunoglobulin-like fold involved in receptor dimerization, we reasoned that *PDGFRA*$\Delta^7$ might enhance PDGF-receptor signaling. We sorted cells by *PDGFRA*$\Delta^7$ expression and identified genes that showed a strong rank correlation with *PDGFRA*$\Delta^7$ levels (Materials and Methods). Positively correlated genes were enriched for the PDGF-receptor signaling network and cell cycle, when compared to the Pathway Commons and DAVID databases (Huang *et al*, 2009; Cerami *et al*, 2011). Negatively correlating genes were enriched for oxidative phosphorylation (Fig 4B). Genes correlating with increasing *PDGFRA*$\Delta^7$ were scanned for overrepresented transcription factor binding motifs. Genetic and physical interaction databases were queried against significant transcription factors (Fig 4C), implicating STAT1 and NF-κB as downstream effectors of *PDGFRA*$\Delta^7$. By comparison, an analogous dose–response analysis of *EGFR* in SF10345 identified an increasing gene set of cell cycle genes, as well as genes related to chromatin modification and cell motion. Inference of mediating transcription factors implicated STAT signaling, as in SF10282. Additionally, SOX2 [a pluripotency factor highly expressed in embryonic, neural and glioma stem cells (Suvà *et al*, 2014)] and c-Jun [an anti-apoptotic factor involved in glioma-genesis (Yoon *et al*, 2012)] targets correlated to *EGFR* dose (Fig 5).

## *PDGFRA*$\Delta^7$ confers a growth advantage and stimulates wild-type *PDGFRA* expression *in vitro*

We expressed *PDGFRA*$\Delta^7$, wild-type *PDGFRA* and a GFP control from lentivirus, in two patient-derived cell lines that we had cultured as monolayers (Fig 6A). One we derived from SF10360 (described

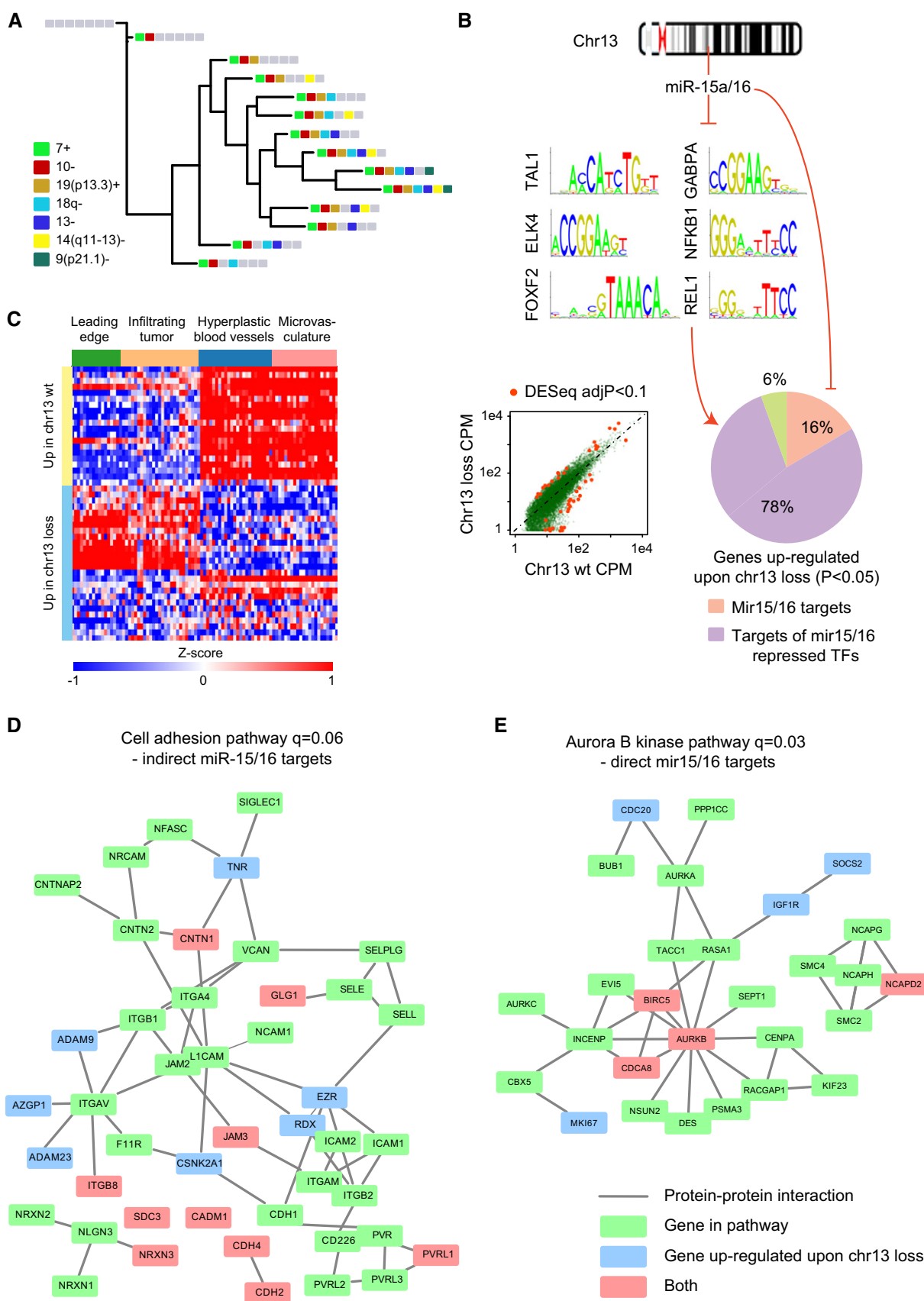

**Figure 3.**

**Figure 3.   Analysis of gene sets differentially expressed between subclones of SF10360.**

A   A phylogeny of CNV cellular genotypes identified in SF10360. Each leaf corresponds to a genotype, defined by a set of CNV presence/absence calls shared between a group of cells.

B   Overrepresented transcription factor recognition motifs in genes up-regulated (DESeq, $P < 0.05$) upon chromosome 13 loss. 16% of up-regulated genes are direct, validated miR-15a/16 targets. Another 78% of up-regulated genes are targets of transcription factors that are repressed by miR-15a/16.

C   Heatmap of the most significantly, differentially expressed genes (DESeq, FDR < 0.1) upon chromosome 13 loss in the Ivy Glioblastoma Atlas. Each row is a gene, each column is an RNA-seq from an anatomically defined tumor compartment, micro-dissected from an untreated GBM biopsy.

D   Protein interaction network of genes differentially expressed upon chromosome 13 loss, which are targets of a miR-15a/16 regulated transcription factor, with an overlay of protein interactions in the cell-adhesion pathway.

E   Protein interaction network of genes differentially expressed upon chromosome 13 loss, which are direct, validated targets of miR-15a/16, with an overlay of protein interactions in the Aurora B kinase.

Data information: Significance of network overlaps in (D, E) was computed via JEPETTO ($q < 0.1$).

here), and the second was from a primary GBM: SF10281. These cell lines do not strongly express *PDGFRA* endogenously, but we detected robust expression of *PDGFRA* and *PDGFRA*$\Delta^7$ mRNA in the respective cultures where they were ectopically expressed (Fig 6B). To specifically quantify the expression of wild-type *PDGFRA,* we designed an RT–qPCR assay with a probe targeted to the deleted region. Intriguingly, we found that endogenous, wild-type *PDGFRA* was induced in both cell lines upon ectopic expression of *PDGFRA*$\Delta^7$ (Fig 6C). When we identified genes that were differentially, recurrently expressed in both *PDGFRA*$\Delta^7$ cultures compared to wild-type *PDGFRA* and GFP, we found that these genes enriched for gene-ontology molecular functions associated with PDGF binding and the binding of other growth factors (Fig 6D). In particular, we saw an up-regulation of the epiregulin encoding mRNA (*EREG*), an epidermal growth factor family member which ligates EGFR and most members of the v-erb-b2 oncogene homolog (ERBB) family. We saw an up-regulation of the notch-receptor ligand jagged 1 (*JAG1*) and the master regulator of angiogenesis, *VEGFA*. Additionally, we saw an induction of regulators of inflammation *IL1B* and *IL6*, as well as *COX-2* and colony-stimulating factor 3 (*CSF3*). *VEGFA*, *COX-2,* and *CSF3* all encode chemotactic factors for MDSC (Lechner *et al*, 2010; Fujita *et al*, 2011; Cao *et al*, 2014; Fig 6E). Additionally, we performed an MTT colorimetric assay and found that ectopic *PDGFRA*$\Delta^7$ expression significantly enhanced cell growth *in vitro*, compared to over-expression of wild-type *PDGFRA* or GFP control (Fig 6F).

## In-frame deletions in the PFGFRA dimerization domain are frequent events in The Cancer Genome Atlas's GBM data

We then processed exome-seq data from 389 GBM patients and corresponding blood controls available from The Cancer Genome Atlas (TCGA) and quantified the frequency of in-frame deletions in *PDGFRA* (Materials and Methods). An in-frame deletion resulting in the loss of exons 8 and 9 (*PDGFRA*$\Delta^{8,9}$) had been previously cloned from a GBM biopsy (Kumabe *et al*, 1992). *PDGFRA*$\Delta^{8,9}$ affects the same dimerization domain as *PDGFRA*$\Delta^7$ and has been shown to be transforming (Clarke & Dirks, 2003). A more recent TCGA study showed that *PDGFRA* mRNA lacking exons 8 and 9 was expressed in 17.8% of GBMs; however, *PDGFRA*$\Delta^{8,9}$ prevalence was not interrogated at the DNA level (Brennan *et al*, 2014). We compared the distributions of reads mapping exons 8 and 9 in *PDGFRA* between tumor samples and the blood controls in TCGA data. The tumor distribution is clearly bimodal (Fig 7A), and this second mode

corresponds to a set of samples depleted of reads mapping exons 8 and 9. By thresholding at the 10% level of the blood distribution as a control, we estimate *PDGFRA*$\Delta^{8,9}$ occurs in 16% of cases in our dataset ($n = 389$), after Benjamini–Hochberg correction for multiple hypothesis testing. This is remarkably close to the 17.8% of cases where *PDGFRA*$\Delta^{8,9}$-consistent mRNA was observed in the TCGA study ($n = 206$). Additionally, we found a second family of deletions in exon 7, occurring in 1.8% of cases we analyzed (including SF10282). The frequency of *PDGFRA* amplification was 13.6%, but we did not find a strong correlation between *PDGFRA*$\Delta^{8,9}$ and *PDGFRA* amplification. On the other hand, all of the small deletions occurred in *PDGFRA* amplified cases (Fig 7B). Both *PDGFRA*$\Delta^7$, *PDGFRA*$\Delta^{8,9}$, and the other small in-frame deletions target immunoglobulin I-set sub-domains of the extracellular domain of PDGFRA (Fig 7C). These domains are involved in receptor dimerization (Chen *et al*, 2012).

## Accumulating mutations correlates with the acquisition of an OPC signature in a proneural GBM and an invasion signature in a classical/mesenchymal case

We performed differential gene expression analysis between SF10282 and SF10345, and as expected, *EGFR* was up-regulated in SF10345 (Dataset EV8). Additionally, there was an overrepresentation of cell-adhesion molecules and genes mediating motility in SF10345's differentially expressed genes (Fig 8A). For example, *CD44*, encoding an adhesion molecule that mediates stem cell homing (Pietras *et al*, 2014), was over 14-fold enriched in SF10345 (Fig 8B). Intriguingly, transcripts coding for chemotactic factors for myeloid-derived suppressor cells (MDSC) were differentially expressed in SF10345. C3 convertase is a core component of the complement cascade, mediating inflammation, and the innate immune response. Complement pathway cytokines attract MDSC and induce their expression of reactive-oxygen species, contributing to a tumor-supportive microenvironment (Markiewski *et al*, 2008). Periostin (POSTN) is secreted by glioma stem cells, recruiting tumor-associated macrophages that enhance tumor growth (Zhou *et al*, 2015). Both *C3* and *POSTN* were up-regulated in SF10345 by several hundred fold (Fig 8B).

On the other hand, the neuron-differentiation pathway was significant in genes up-regulated in SF10282 (Fig 8A). Upon inspection, however, these genes were factors predominantly expressed by OPCs during development: *PDGFRA*, *NKX2-2*, *SOX10*, *SEMA5A*, *LINGO1*, *S100B*, *MAP2* (Shafit-Zagardo *et al*, 2000; Deloulme *et al*,

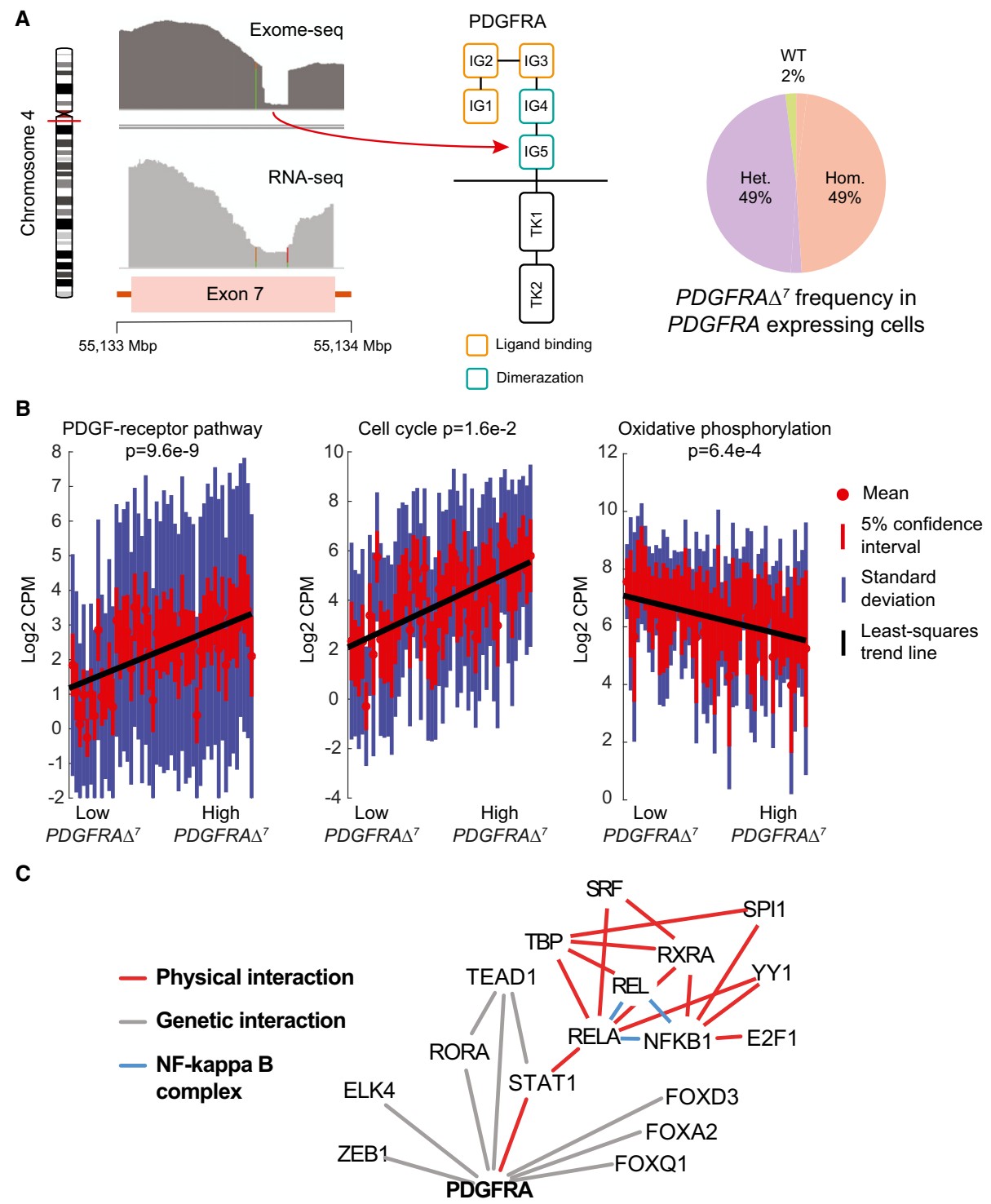

**Figure 4.  Dose–response analysis of a *PDGFR* mutant.**

A   Coverage of exome-seq (top left) and RNA-seq reads (bottom left) in exon 7 of the *PDGFRA* gene. The deletion targets the immunoglobulin-like domain IG5 of the PDFG receptor (center). 49% of *PDGFRA* expressing cells express *PDGFRAΔ⁷* homozygously, another 49% express it heterozygously (right).

B   Enriched gene sets (WEBGESTALT, DAVID, adj. *P*-value < 0.05) correlated to *PDGFRAΔ⁷*. Distributions of in-pathway genes in individual cells, sorted from low *PDGFRAΔ⁷* to high *PDGFRAΔ⁷*.

C   An interaction network (generated via geneMANIA) of physical and genetic interactions of transcription factors, whose recognition motifs are overrepresented (OPOSSUM, *z*-score ≥ 10, Fisher score ≥ 7) in correlated genes. Physical interactions are interactions between the protein products, identified from proteomics experiments. Genetic interactions are changes in gene expression that occur when another gene is suppressed in a knockdown experiment.

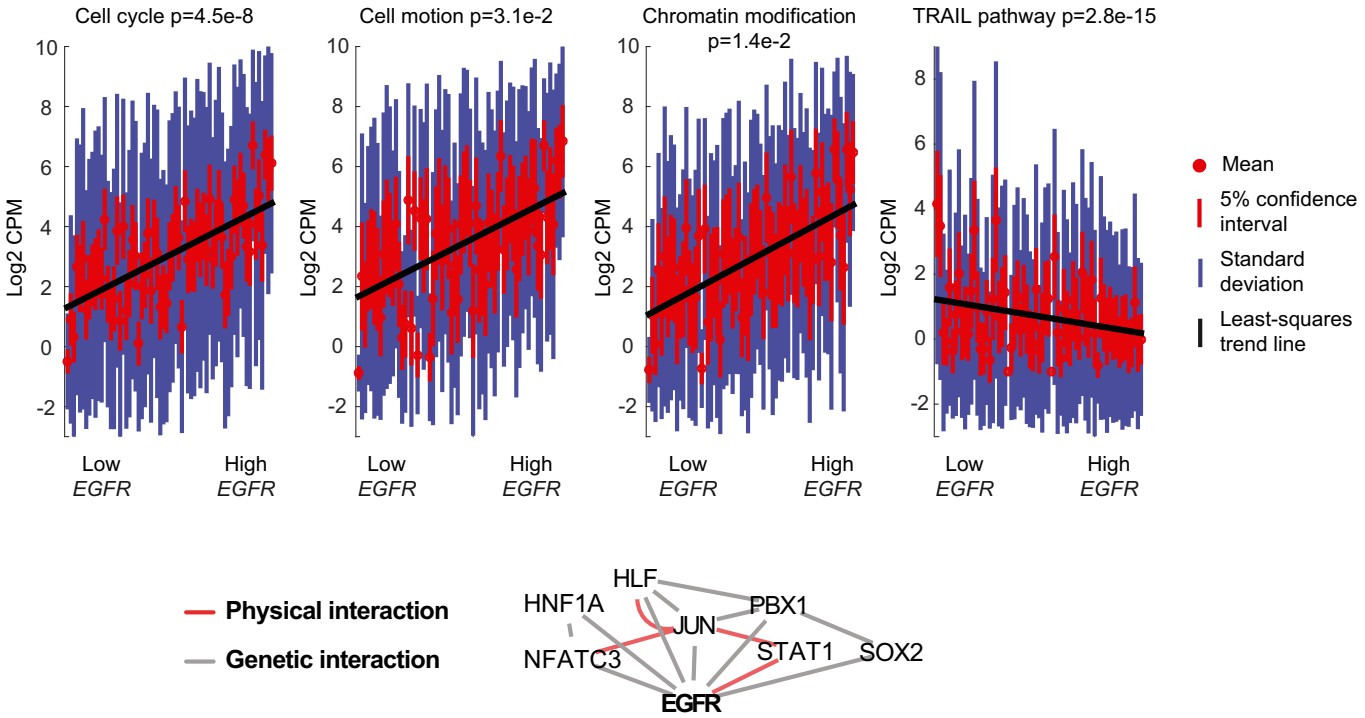

**Figure 5.  Dose–response analysis of an *EGFR*-amplified case.**
Enriched gene sets (WEBGESTALT, DAVID, adj. *P*-value < 0.05) correlated to *EGFR*. Distributions of in-pathway genes in individual cells, sorted from low *EGFR* to high *EGFR* (top panel). An interaction network (generated via geneMANIA) of physical and genetic interactions of transcription factors, whose recognition motifs are overrepresented (OPOSSUM, *z*-score ≥ 10, Fisher score ≥ 7) in correlated genes. Physical interactions are interactions between the protein product, identified from proteomics experiments. Genetic interactions are changes in gene expression that occur when another gene is suppressed in a knockdown experiment.

2004; Goldberg, 2004; Petryniak *et al*, 2007; Jepson *et al*, 2012). Our initial inspection of SF10282 had revealed a NSC-like subpopulation (Fig 1C), which occurred early in our phylogeny of SF10282. Analysis of NSC and OPC gene expression in our phylogeny showed constitutively high expression of *PAX6*, *SOX2,* and *TNC*, but a gradual increase in the OPC genes *OLIG2*, *ASCL1*, *NKX2-2,* and *SOX10* along pseudo-time. PI3K/AKT pathway genes and genes implicated in angiogenesis also increased concomitantly (Fig 8C). By comparison, SF10345 showed a progressive up-regulation of genes encoding extracellular matrix and transmembrane proteins associated with glioma motility and invasion, such as tenacin-C, neurocan, and integrin (Cuddapah *et al*, 2014). AKT pathway genes *AKT2* and *AKT3*, which contribute to glioma invasiveness and malignancy

(Chautard *et al*, 2014), and class II myosins, required for glioma invasion and neural stem cell migration (Beadle *et al*, 2008; Ostrem *et al*, 2014), were also progressively up-regulated as one moves along the backbone of the SF10345 phylogeny (Fig 8D).

# Discussion

Despite standard of care treatment, GBM has an extremely high recurrence rate, approximately 90% (Weller *et al*, 2013). There is an urgent need for combinatorial strategies to address residual disease (Prados *et al*, 2015). In particular, intra-tumor receptor heterogeneity is a confounder for tyrosine-kinase inhibitor therapy

**Figure 6.  *In vitro* analysis of *PDGFRA*Δ⁷.**

A   Schematic representation of the *in vitro* experiment.
B   Reads mapped to exon 7 of *PDGFRA* in *PDGFRA*Δ⁷ over-expressing, wild-type *PDGFRA* over-expressing, and GFP control cultures.
C   Quantitative PCR with a probe targeted to the region deleted in *PDGFRA*Δ⁷. Results (mean ± SD) comparing wild-type *PDGFRA* expression between GFP control and *PDGFRA*Δ⁷ expressing cells from SF10360. The asterisk indicates *P* < 0.05 (*t*-test).
D   Top: Volcano plot of gene expression between *PDGFRA*Δ⁷ and *PDGFRA* wild-type expressing cells from SF10281 (left) and SF10360 (right). Differentially expressed genes (adjusted *P*-value < 0.05, ANODEV test from DEGSeq2 package) are indicated in red. Bottom: Gene-ontology enrichment of genes differentially expressed between *PDGFRA*Δ⁷ and *PDGFRA* wild type in both cell lines (right).
E   Bar plots of mean gene expression (± SD) across duplicates in GFP, wild-type PDGFRA, and expressing cells from SF10281 (left) and SF10360 (right).
F   WST-1 assay (*n* = 3) comparing proliferation (mean ± SD) of SF10360c cells expressing GFP, PDGFRA WT or PDGFRA delta7. The asterisk indicates *P* < 0.05 (*t*-test).
G   Genetic interactions and physical interactions (via Genemania) of transcription factors (via OPOSSUM) whose motifs are enriched in the promoters of genes correlating with *PDGFRA*Δ⁷ *in vivo*. This is compared to the genetic interactions between upstream transcription factors of genes which are differentially expressed in the *PDGFRA*Δ⁷ over-expression experiment.

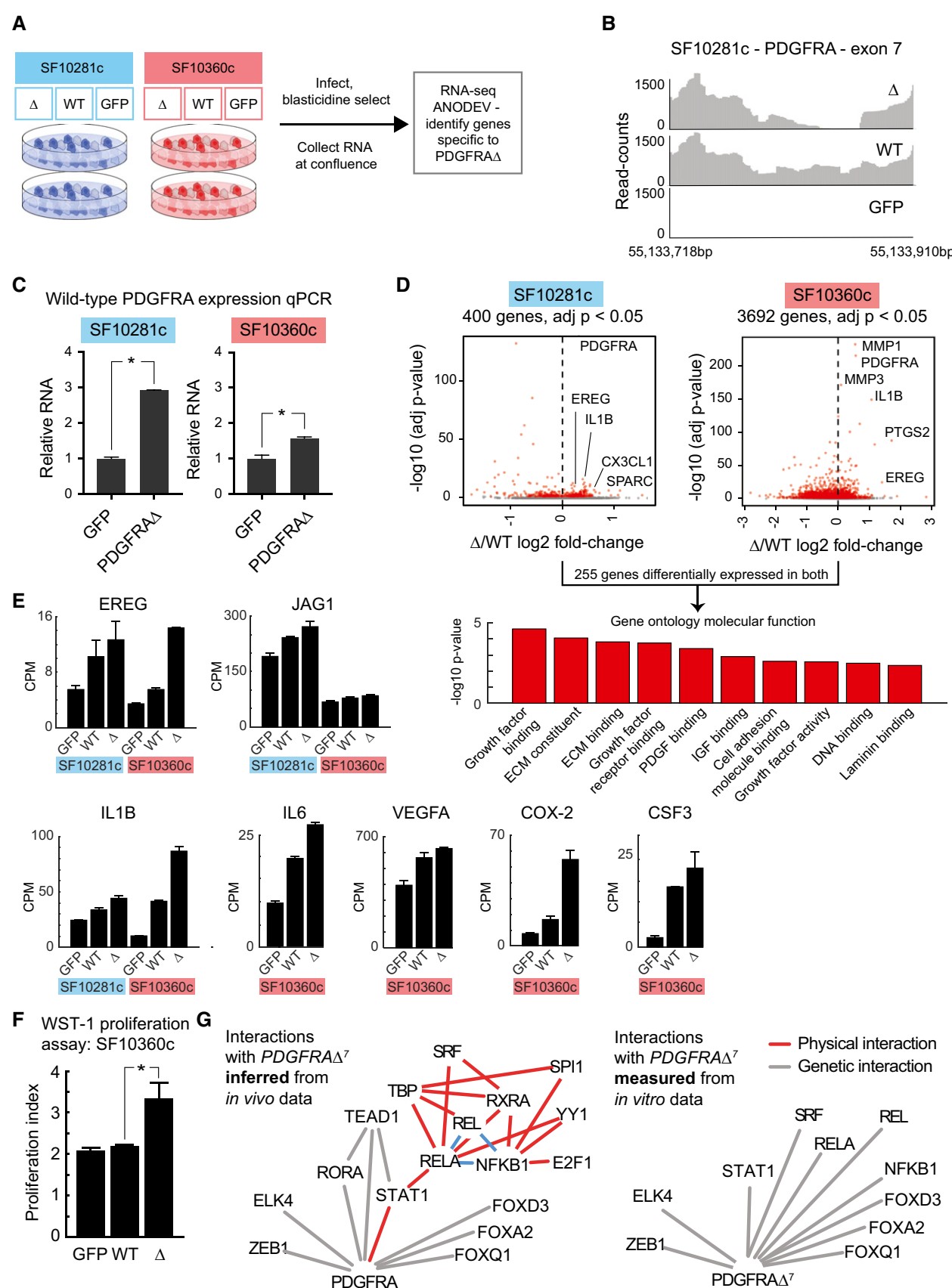

Figure 6.

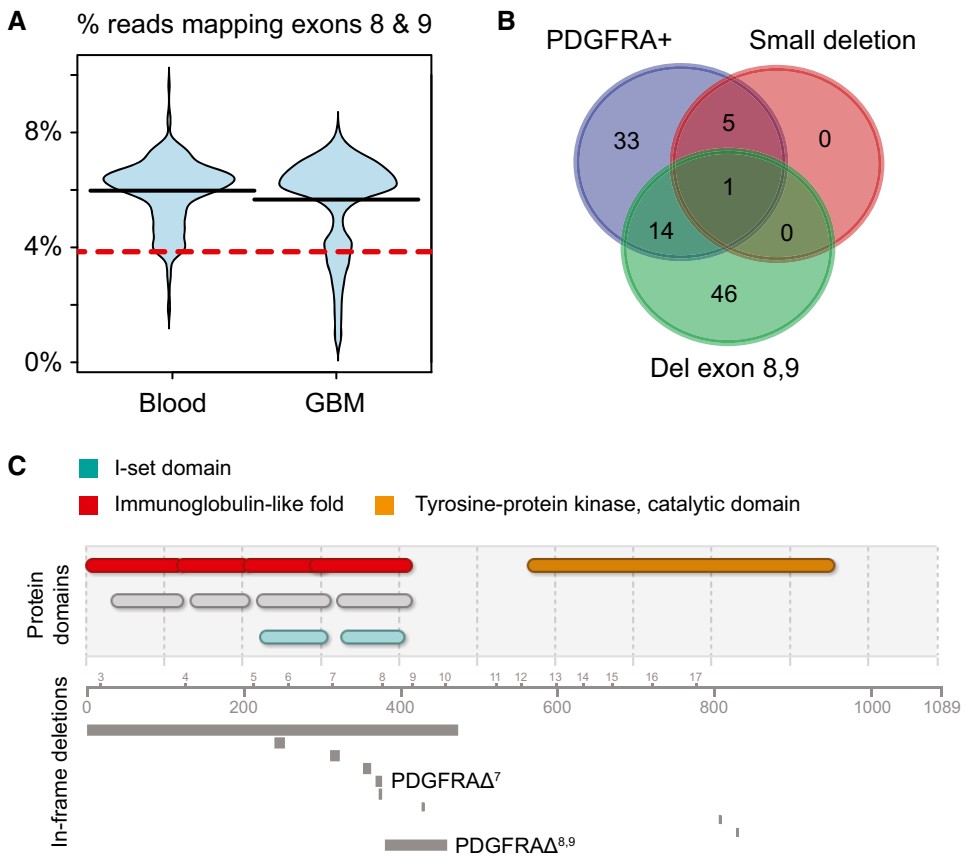

**Figure 7.  Analysis of TCGA data reveals a family of *PDGFRA* mutations affecting the dimerization domain.**

A   The fraction of reads assigned to exons 8 and 9 from all exome-seq reads mapping to *PDGFRA*, compared between blood control and GBM samples. The 10[th] percentile of the blood control distribution is indicated by a red line.

B   Venn diagram of GBM cases harboring an amplification, a deletion of exons 8 and 9, or a small deletion affecting the dimerization domain of PDGFRA.

C   Visualization of deletions detected in *PDGFRA* from TCGA data. Domains are indicated by color (top), exons and protein residues by number (middle), and deleted regions by bars (bottom).

(De Witt Hamer, 2010). Recent advances in single-cell analysis have enabled high-resolution estimates of clonal heterogeneity (Francis *et al*, 2014; Patel *et al*, 2014; Meyer *et al*, 2015). In this study, we combined whole exome and single-cell mRNA sequencing to map transcriptional signatures to mutational phylogenies in EGF- and PDGF-driven GBMs. These data implicate in-frame deletions in the dimerization domain of PDGFRA as potential therapeutic targets. And, they identify a cell-type hierarchy, which occurs in early brain development, as being recapitulated during tumor evolution.

In the developing forebrain, OPCs arise from neuroepithelial stem cells in sequential waves of oligodendrocyte production (Kessaris *et al*, 2006; Menn *et al*, 2006). ASCL1 and OLIG2 are

interacting transcription factors required for oligodendrogenesis and highly expressed in OPCs (Zhou & Anderson, 2002; Petryniak *et al*, 2007; Nakatani *et al*, 2013). OLIG2 drives SOX10 transcription in OPCs (Kuspert *et al*, 2011), which in turn regulates myelin expression (Stolt *et al*, 2002), critical for oligodendrocyte function. Consistent with aberrant activation of this developmental program in GBM, SF10282 progressively up-regulated the above OPC-specific genes in its model of tumor evolution. Early cells expressed *PAX6*, *SOX2*, and other markers of neural stem cells but produced daughters with a more OPC-like profile (Fig 8C). *PDGFRA* expression is characteristic of OPCs in non-malignant brain and was amplified in SF10282.

**Figure 8.  Differences in gene expression between an EGF- and a PDGF-driven GBM.**

A   A scatterplot of log2 mean expression between SF10345 and SF10282. Adjusted *P*-values of biological process GO terms, enriched in differentially expressed genes (SCDE), with an adjusted *P*-value < 0.05 (on a −log10 scale).

B   Single-cell gene expression estimates (colored lines), their SF10345/SF10282 log2 fold-changes, and their joint posteriors for select genes. A 95% confidence interval is represented by dotted lines.

C, D   Phylogenetic trees for SF10282 and SF10345. Each leaf corresponds to a unique set of cells with the same CNV genotype. Bar plots show mean (± SD) expression of select genes across sets of cells that progressively gain CNVs.

▶

    

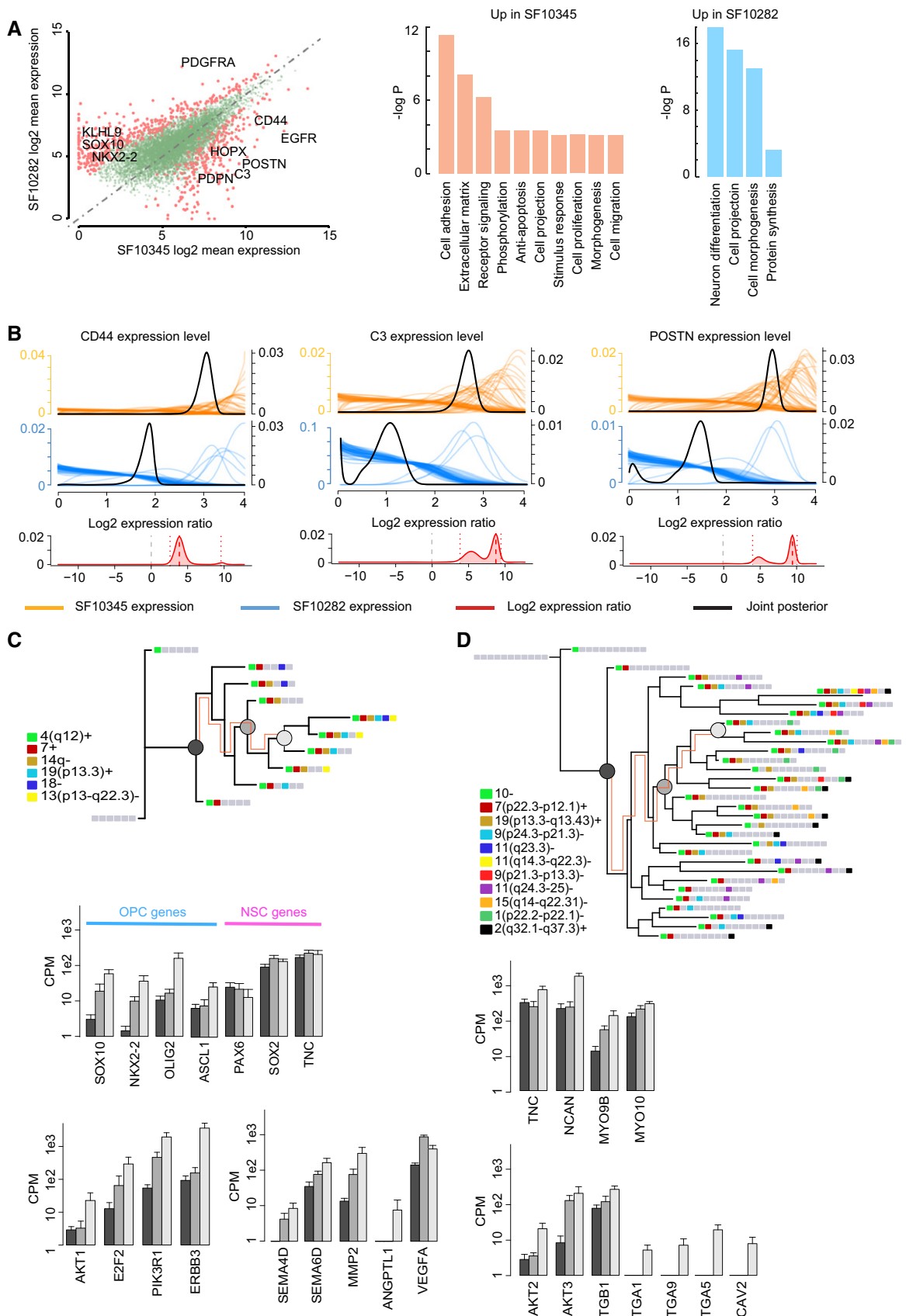

**Figure 8.**

However, 68% of tumor cells in SF10282 express an in-frame deletion mutant, $PDGFRA\Delta^7$. We found that $PDGFRA\Delta^7$ is part of a family of in-frame deletions in $PDGFRA$'s dimerization domain that occur in GBM. The most common of these mutations, $PDGFRA\Delta^{8,9}$, occurs with 16% frequency in the TCGA DNA sequencing data we analyzed ($n = 389$). Our finding is consistent with a recent TCGA paper that found $PDGFRA$ mRNA lacking exons 8 and 9 in 17.8% of their samples ($n = 206$; Brennan *et al*, 2014). $PDGFRA\Delta^{8,9}$ expression has been shown to induce constitutive signaling of the PDGF receptor and is sufficient for malignant transformation (Clarke & Dirks, 2003). In-frame deletions in $PDGFRA$'s dimerization domain have also been observed in pediatric high grade gliomas, and those studies implicate constitutive receptor signaling too (Paugh *et al*, 2013). Our single-cell data show that increasing $PDGFRA\Delta^7$ dosage correlates *in vivo* with an activation of genes downstream of the PDGF-receptor, in particular, targets of rapid-acting transcription factors (e.g. STAT, NF-κB). When ectopically over-expressed, in patient-derived glioma cell lines, $PDGFRA\Delta^7$ enhances proliferation compared to wild-type $PDGFRA$ over-expression. $PDGFRA\Delta^7$ induces wild-type $PDGFRA$ expression *in vitro*, along with tyrosine-kinase receptor ligands and pro-angiogenesis genes.

The deletions we observed aggregate in the I-set domains of $PDGFRA$'s immunoglobulin-like folds. These highly conserved domains are involved with receptor dimerization and in principle can interact with a variety of different domain types; however, these domains are typically glycosylated, which protects against promiscuous receptor interaction in the absence of ligand (Barclay, 2003). All of the deletions we have observed are either on or near predicted glycosylation sites (Fig EV3).

Single-cell RNA-seq has enabled composition assessments and lineage reconstruction in the highly dynamic, highly heterogeneous context of the developing brain (Darmanis *et al*, 2015; Pollen *et al*, 2015; Diaz *et al*, 2016; Liu *et al*, 2016). Single-cell RNA-seq of tumor biopsies provide a similar snapshot of tumor evolution. By using a proven approach such as bulk exome-seq to identify mutations, single-cell RNA-seq libraries can then be triaged to assign a transcriptional signature to a mutational profile or phylogeny. By cross-referencing public atlases such as the Ivy Glioblastoma Atlas, we further related this information to tumor anatomical structure. When we cross-referenced the Ivy Glioblastoma Atlas with the chromosome 13 deletion subclone of SF10360, we identified a spatial segregation of differentially expressed genes, aggregating in the perivascular niche and the leading edge, respectively. Cells harboring the chromosome 13 deletion were also characterized by an up-regulation of pro-invasion, adhesion genes. However, whether the deletion event preceded or succeeded any physical separation of these two clones could not be inferred from these data. Targeted resections that record relative biopsy position can resolve spatial evolution more accurately (Sottoriva *et al*, 2013).

## Materials and Methods

### Sample acquisition and processing

Fresh tumor tissue was acquired from patients undergoing resection for glioblastoma. De-identified samples were obtained from the Neurosurgery Tissue Bank at the University of California San Francisco (UCSF). Sample use was approved by the Committee on Human Research at UCSF. The experiments conformed to the principles set out in the WMA Declaration of Helsinki and the Department of Health and Human Services Belmont Report. All patients provided informed written consent. Tissues were minced in collection media (Leibovitz's L-15 medium, 4 mg/ml glucose, 100 μg/ml penicillin, 100 μg/ml streptomycin) with a scalpel. Samples were further dissociated in a mixture of papain (Worthington Biochem. Corp) and 2,000 units/ml of DNase I freshly diluted in EBSS and incubated at 37°C for 30 min. The suspension was centrifuged for 5 min at 300 *g* and re-suspended in PBS. Suspensions were triturated by pipetting up and down 10 times and passed through a 70-μm strainer cap (BD Falcon), followed by centrifugation for 5 min at 300 *g*. Pellets were re-suspended in PBS and passed through a 40-μm strainer cap (BD Falcon), followed by centrifugation for 5 min at 300 *g*. Dissociated, single cells were then re-suspended in GNS (Neurocult NS-A (Stem Cell Tech.), 2 mM L-glutamine, 100 U/ml penicillin, 100 μg/ml streptomycin, N2/B27 supplement (Invitrogen), sodium pyruvate).

Single-cell capture and cDNA generation was performed using the Fluidigm C1 Single-Cell Integrated Fluidic Circuit (IFC) and SMARTer Ultra Low RNA Kit. cDNA was quantified using Agilent High Sensitivity DNA Kits and diluted to 0.15–0.30 ng/μl. Dual indexing and amplification were performed using the Nextera XT DNA Library Prep Kit (Illumina) according to the Fluidigm C1 protocol. 96 single-cell RNA-seq libraries were generated from each tumor sample and were pooled for 96-plex sequencing. Amplified and pooled cDNA was purified and size selected twice using 0.9× volume of Agencourt AMPure XP beads (Beckman Coulter). Final cDNA libraries were quantified using High Sensitivity DNA Kits (Agilent) and sequenced on a HiSeq 2500 (Illumina), using the paired-end 100 base pair (bp) protocol.

### Exome-sequencing and genomic mutation identification

Exome capture was done using NimbleGen SeqCap EZ Human Exome Kit v3.0 (Roche) exome capture kits on a tumor sample and a blood control sample from each patient. Sequencing was carried out with an Illumina-HiSeq 2500 machine acquiring 100-bp paired-end reads. Reads were aligned to the human genome (hg19) using BWA (Li & Durbin, 2009), whereas only uniquely matched paired reads were retained. PicardTools (http://broadinstitute.github.io/picard/) and the GATK toolkit (McKenna *et al*, 2010) were used for quality score re-calibration, duplicate-removal, and re-alignment around indels. The resulting BAM files were sorted by genomic coordinates. Subsequently, the percent contamination was assessed with ContEst (SF10345: 0.1%, SF10360: 0.1%, SF10282: 0.2%). OxoG metrics were calculated with PicardTools' CollectOxoGMetrics (Dataset EV9). After these control steps, single nucleotide variants (SNVs), short indels (< 50 bps), and large CNVs comprising more than three exons were detected. Somatic SNVs were inferred with MuTect (https://www.broadinstitute.org/cancer/cga/mutect) for each tumor/control pair and annotated with. SNVs with < 10% variant frequency in the tumor, with more than five variant reads in the patient-matched normal, or > 10% variant frequency in the patient-matched normal were excluded from further analysis. Small indels were detected with Pindel (Ye *et al*, 2009) and those with fewer than six supporting reads in the tumor, any supporting reads or

   

< 14 total reads in the patient-matched normal, and replacements for which the deletion and non-template inserted sequence were of the same length were excluded. All indels and SNVs were annotated for their mutational context and effect using the Annovar software package (Wang *et al*, 2010). Only protein-coding or splice-site mutations were retained for further analysis. ADTEx (Amarasinghe *et al*, 2014) was used for detection of large somatic CNVs. Only CNVs comprising more than 100 exons were retained for downstream analysis. Proximal (< 1 Mbp) somatic CNVs were merged in the output file to maximize CNV regions.

**Single-cell RNA-sequencing data preprocessing, quality control, and GBM subtype classification**

Reads were trimmed for quality and Nextera adapters removed with TrimGalore! (http://www.bioinformatics.babraham.ac.uk/projects/trim_galore/), and paired-end reads were mapped to the human genome (hg19) with tophat2 (Kim *et al*, 2013) using the –prefilter-multihits option and a GENCODE V19 transcriptome-guided alignment. Quantification of GENCODE genes was carried out with featureCounts (Liao *et al*, 2014). Only fragments corresponding to uniquely mapped, correctly paired reads were kept. Expression values were normalized to CPM in each cell. We filtered samples whose background fraction is significantly high, via a threshold on the (Benjamini–Hochberg corrected) $q$-value of a Lorenz statistic on the samples' cumulative densities, described previously (Diaz *et al*, 2016). In our tests, samples that have a small $q$-value have low complexity, as measured by Gini-Simpson index, and they have low coverage, as estimated by the Good–Turing statistic (Fig EV1).

To compare genotypes from the single-cell RNA-seq data across patients and to the exome-seq data, we identified 17, 21, and 15 single nucleotide variants (SNVs) in the single-cell RNA-seq that are patient specific in SF10282, SF10345, and SF10360, respectively. These SNVs are likewise detected in, and only in, their respective blood and tumor-derived exome-seq datasets. The median difference in RNA-seq to tumor exome-seq variant allele frequency (VAF) is 0.056. By comparison, the median difference in tumor exome-seq to blood exome-seq VAF is 0.044 (see Appendix Fig S3 and Appendix Table S1).

Infiltrating stromal/non-malignant cells were identified as those cells which contained none of the CNVs, none of the indels or point mutations found in the exome-seq data, and clustered away from tumor cells in a hierarchical clustering. These putative stromal cells formed two clusters: one that is comprised of cells from all samples that expresses markers of immune cells (particularly those of macrophages/microglia) and a second cluster comprised of cells from SF10282 that express oligodendrocyte markers (Fig EV2C). We classified all of the remaining cells according to the Verhaak *et al* (2010) and Sun *et al* (2014) molecular subtypes. To perform the Verhaak classification in individual cells, we fit a linear regression model using the four centroids in the original Verhaak clustering as predictors and the gene expression profile of the tumor cell to be classified as a response. We restricted expression profiles in individual cells to those genes used in the original Verhaak clustering and represented expression by standardized, log-transformed CPMs. The cell to be classified is assigned to the subtype whose corresponding centroid has the largest regression coefficient in magnitude. Sun *et al* determined their subtypes by identifying gene modules that

co-expressed with *PDGFRA* or *EGFR* in a large cohort of adult diffuse gliomas. In each cell to be classified, we averaged gene expression (measured by log-CPM) across both of these two gene modules. If either module's average was more than twofold higher than the other, then we assigned the cell to the corresponding subtype.

**Single-cell CNV presence/absence calls**

Based on the premise that copy-number changes are reflected in RNA-seq read counts when averaged over large, adjacent genomic regions (Patel *et al*, 2014), we examined loci that were called for somatic CNV in our ADTEx pipeline. For each CNV candidate region (CNVCR), we sum the library-size normalized read counts across genes in that region, for each cell in our tumor sample. We do the same for each cell in a non-malignant, human brain control (Darmanis *et al*, 2015). We use the distribution across cells in the control for each CNVCR, to assess the sum in a given tumor cell. We use the 5% significance level of the control distribution as our threshold for making a CNV call (Fig 2A), and control for multiple hypothesis testing using Benjamini–Hochberg correction. This results in a genotype assignment to each cell, determined by which CNVs called in the exome-seq data are present in that cell. To estimate the false discovery rate (FDR) of this classification procedure, we performed 10-fold cross-validation using the normal-brain control cells. For each patient, we randomly selected tranches of 10% as test and 90% as training data. We estimated the FDR as (# positive CNV calls)/(# total CNV calls), for each of the 10 folds. We found the FDR to be < 0.01 for all tests (Appendix Fig S2A). As a second estimator of the FDR, we classified the presence of CNVs on a dataset comprised of non-malignant, fetal-brain cells (Pollen *et al*, 2015), and estimated the FDR as above. We found these FDR estimates to all be < 0.06 (Appendix Fig S2B).

**Phylogenetic trees**

We measure pairwise distance between individual cells using Jaccard distance between CNV genotypes. This measures the number of shared CNV calls, as a fraction of the number of unique calls in either cell. To obtain a phylogenetic tree of tumor cells based on this distance metric, we use the Fitch–Margoliash method (Fitch & Margoliash, 1967) as implemented in the phylip R package (Revell & Chamberlain, 2014), adding a "normal" genotype with no CNVs as an out-group to root the tree. We identified 5–6 cells per sample that did not harbor the CNV with the highest frequency (chromosome 7 gain in SF10345, chromosome 4q12 gain in SF10282 and chromosome 10 loss in SF10360). Since these mutations are nearly ubiquitous, they represent founding mutations for the dominant clones sampled in our cells. This is consistent with the fact that they affect cancer driver genes *EGFR*, *PDGFRA,* and *PTEN,* respectively. These rare cells lacking founding mutations may be technical outliers, or members of a lineage under-sampled in the biopsy.

**PDGFR *in vitro* overexpression**

Primary dissociated tumors were plated for cell culture in DMEM-F12 with 10% FBS. Expression vectors driving wild-type *PDGFR*, *PDGFRΔ[7]*, or GFP (pLV[Exp]-EGFP/Bsd-EF1A) were generated and

packaged into third generation lentivirus particles (VectorBuilder, Cyagen Biosciences). Triplicate cell cultures were infected with lentiviruses at equal titers at a MOI of ~1.0 and with 0.5 μg/ml polybrene. Two days following infection, cells were selected with 33.3 μg/ml blasticidine for 3 days. For RNA collection, cells were grown for one additional day with no drug selection and then harvested. For proliferation assays, cells were grown continually in the presence of 33.3 μg/ml blasticidine. Vector expression was confirmed using fluorescence microscopy and RNA-seq.

### Bulk RNA-seq sample processing

RNA was harvested using TRIzol reagent, followed by Direct-zol MiniPrep RNA purification kits (Zymo Research) with the on-column DNase digestion step. RNA integrity was confirmed using the Agilent 2200 RNA ScreenTape (Agilent Technologies). RNA-seq libraries were generated using TruSeq Stranded mRNA kit according to manufacturer's protocol (Illumina). cDNA was validated using the Agilent 2200 DNA 1000 ScreenTape, Qubit 2.0 Fluorometer (Life Technologies), and ddPCR (Bio-Rad). Cluster generation and sequencing was performed on a HiSeq 4000, using the single-end 50 read protocol.

Reads were mapped to the human genome (hg19) with tophat2 (Kim *et al*, 2013). Quantification of GENCODE V19 genes was carried out with featureCounts (Liao *et al*, 2014) using only uniquely mapped reads. Differential expression analyses were performed via DESeq2, using the likelihood ratio test applied against the wild-type *PDGFR*, *PDGFR*$\Delta^7$, and GFP triplicate samples as a three-level factor.

### Dose–response analysis

In SF10282, cells were sorted by *PDGFRA*$\Delta^7$ expression, in SF10345 cells were sorted by *EGFR* expression. Genes which had a Spearman's rank correlation in the top 5% with *PDGFRA*$\Delta^7$ were considered for further analysis. Pathway analysis was done using Pathway Commons (Cerami *et al*, 2011) via WEBGESTALT (Wang *et al*, 2013), and DAVID (Huang *et al*, 2009), transcription factor motif enrichment analysis was done via OPOSSOM (Sui Ho *et al*, 2007). Network interactions were computed via GeneMANIA (Montojo *et al*, 2010).

### *In vitro* proliferation and wild-type *PDGFRA* quantification assays

For the WST-1 proliferation assay, $1 \times 10^4$ cells were cultured in a 96-well plate for 24 h in 100 μl of complete media. Then, 10 μl of WST-1 reagent (Roche) was added to each well. Cells were incubated at 37°C, 5% $CO_2$ for 4 h, and placed on a shaker for 1 min. The plates were then read on a microplate reader with a wavelength of 420 nm and a reference at 620 nm. For the TaqMan gene expression assay, cDNA was synthesized from 50 ng of glioma cell line-derived RNA with qScript™ XLT cDNA SuperMix (Quanta Biosciences, Gaithersburg, MD). 2 μl of converted cDNA was then used in the qRT–PCR reaction with PerfeCta® FastMix® II (Quanta Biosciences), according to the manufacturer's protocol. A custom TaqMan assay specific to wild-type, but not mutant, *PDGFRA* was designed by Life Technologies. The fluorescent probe was targeted to the region deleted in *PDGFRA*$\Delta^7$, but present in wild-type PDGFRA. Real-time

PCR and data analysis were performed using the StepOnePlus Real-Time PCR system (Life Technologies, Carlsbad, CA). GAPDH (Assay ID: Hs02758991_g1) expression was used as the housekeeping gene and relative expression was determined using the $2^{\Delta\Delta C_T}$ method.

### Differential expression and time-series analysis

To identify genes differentially expressed between SF10345 and SF10282 and assess inter-cellular heterogeneity, we used the scde R package (Kharchenko *et al*, 2014). We used DESeq to compare chromosome 13 loss to wild-type cells in SF10360 and treated each cell as a replicate. Genes that were expressed in more than 80% of cells at < 1 CPM were filtered prior to each analysis. Transcription factor motif enrichment analysis was done via OPOSSOM (Sui Ho *et al*, 2007). JEPETTO (Winterhalter *et al*, 2014), run via Cytoscape (Shannon *et al*, 2003), was used to compute pathways having significant overlap with genes up-regulated upon chromosome 13 loss, and their protein interactions. For the time-series analysis, we chose paths in our phylogenetic trees corresponding to the most frequent mutation at each level. Our goal was to identify an ordering of the dominant clones in the sample. But, in principle, expression along an arbitrary path can be measured. We grouped cells with identical copy-number profiles into three intermediate points along each branch: early, mid, and late. We subjected monotonically increasing genes to gene-ontology analysis via DAVID (Huang *et al*, 2009).

### TCGA data analysis

Alignments for all GBM exome-seq with available paired blood controls from TCGA (http://cancergenome.nih.gov/) were retrieved from CGHub (https://cghub.ucsc.edu/) in BAM format. Reads in FASTQ format were extracted with bedtools from these alignments (Quinlan & Hall, 2010). We detected the quality encoding with a custom perl script and clipped adapters/low quality based with Trimmomatic (Bolger *et al*, 2014). Next, we mapped reads to the human genome (hg19) with HISAT2 (Kim *et al*, 2015). Small Indels were detected with pindel (Ye *et al*, 2009) for each tumor/control pair. If multiple sequencing libraries existed for one patient, the most recently published library was used. For the detection of loss of exons 8/9, we calculated the fraction of reads mapping to these exons from all reads aligned to the PDGFRA in each GBM and each blood sample. We used the distribution of all blood controls to assign significance values to each GBM sample. *P*-values were corrected for multiple testing with Benjamini–Hochberg (Benjamini & Hochberg, 1995) and an adjusted *P*-value of < 0.25 was considered significant, which is in agreement with the significance threshold for CNVs used by the TCGA consortium (Brennan *et al*, 2014).

### Data availability and algorithm parameters

The RNA-seq and exome-seq data from this publication have been deposited at the European Genome-phenome Archive (EGA, http://www.ebi.ac.uk/ega/) which is hosted at the EBI, under accession number EGAS00001001900. Parameters of all algorithms are available in the Appendix Supplementary Methods.

**Expanded View** for this article is available online.

## Acknowledgements

We would like to thank Joanna Phillips and Anny Shai of the UCSF Neurosurgery Tissue Core, who facilitated tissue acquisition, and Pamela Paris (UCSF) for consultation on genomic arrays. This work has been supported by a Shurl and Kay Curci Foundation Research Grant, a UCSF Brain Tumor SPORE Career Development Award (P50-CA097257-13:7017), and a gift from the Dabbiere Family to A.D., a Damon Runyon Cancer Research Foundation post-doctoral fellowship (DRG-2166-13) to A.P., NIH award 1R01NS091544-01A1, VA award 5I01 BX000252, and gifts from the Hana Jabsheh Initiative and the Dabbiere Family to D.A.L., and by NIH awards U01 MH105989 and R01NS075998 to A.R.K.

## Author contributions

AD conceived of and designed the study. SM developed the algorithms for inferring tumor evolution. ED, GK, SJL, MM, TJN, and AAP performed the experiments under the supervision of AD, ARK, and DAL. MA provided the surgical specimens via the UCSF Tissue Core. AD and SM performed the bioinformatics analysis of single-cell and exome-seq data. AD and SJL performed the bioinformatics analysis of the *in vitro* RNA-seq data. AD and SM wrote the manuscript with input from all authors. All authors read and approved the final manuscript.

## Conflict of interest

The authors declare that they have no conflict of interest.

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
