## [Review Process File · Molecular Systems Biology]

Single-cell sequencing maps gene expression to mutational phylogenies in PDGF and EGF driven gliomas

Sören Müller, S. John Liu, Elizabeth Di Lullo, Martina Malatesta, Alex A. Pollen, Tomasz J. Nowakowski, Gary Kohanbash, Manish Aghi, Arnold R. Kriegstein, Daniel Lim and Aaron Diaz

*Corresponding authors: Aaron Diaz, University of California, San Francisco
Daniel Lim, University of California, San Francisco*

Review timeline:

Submission date:	22 March 2016
Editorial Decision:	20 April 2016
Revision received:	19 July 2016
Editorial Decision:	03 August 2016
Revision received:	16 August 2016
Editorial Decision:	21 September 2016
Appeal received:	04 October 2016
Editorial Decision:	04 November 2016
Revision received:	08 November 2016
Accepted:	20 September 2016

Editor: Maria Polychronidou

Transaction Report:

1st Editorial Decision

20 April 2016

Thank you again for submitting your work to Molecular Systems Biology. We have now heard back from the three referees who agreed to evaluate your study. As you will see below, overall the reviewers think that study is a potentially valuable contribution to the field. However, they raise a number of concerns, which should be carefully addressed in a revision of the manuscript.

The reviewers' recommendations are rather clear so there is no need to repeat all the points listed below. Some of the more fundamental issues are the following:

- Further functional support needs to be provided for the main conclusions.
- The computational and statistical analyses need to be described in better detail so that the reviewers can better evaluate them.

Reviewer #1:

Here, Muller and colleagues perform bulk exome sequencing and single cell mRNA seq analysis of 3 GBM surgical samples. From the exome sequencing and mRNA data, they infer a phylogeny of mutational events and identify the transcriptional consequences of these genetic alterations and model transcriptional kinetics. They identify a PDGF receptor extracellular domain deletion and infer its transcriptional consequences. They also infer a dose dependence of mutations in amplified versions of EGFR and PDGFR on transcriptional programs, and link the PDGFR-driven one with an OPG like phenotype, and the EGFR one with an invasion phenotype.

Overall, the paper is quite interesting and a valuable contribution. The experiments are well thought out, the paper is clearly written, and for the most part, the conclusions are supported by the data. Please see below for suggestions on how to strengthen this aspect. The paper's chief strength lies in the integration of whole exome sequencing and single cell RNA seq of clinical GBM samples.

The paper would be greatly improved by addressing a few issues.

- 1) Biological validation - the manuscript is entirely bioinformatic. This is not per se a problem, but the paper aims to make functional conclusions, not only correlations, for example, that the new PDGFR mutation is gain of function. Even if a similar deletion in the same domain has been previously shown to promote dimerization, it would greatly strengthen the paper to show that this mutation is gain of function. Similarly, the dose dependence of specific transcriptional programs on the mutations is a fascinating inference. However, to conclude this with confidence, it needs to be directly tested by examining gene expression, or at least the level of a few key transcripts, in response to different levels of the key EGFR and PDGFR mutations.
- 2) Additional relevant literature should be cited, included Francis et al, who show construction of phylogenies from single cell DNA seq in GBM.

Reviewer #2:

The authors performed bulk exome sequencing and single-cell mRNA sequencing of human GBM specimens in order to assess the effect of intra-tumor heterogeneity on gene expression. By using this approach they identified a PDGFRA mutant whose dose correlates to an up-regulation of PI3K/Akt pathway gene expression. The approach used is challenging and gives conclusions supporting the effect of intra-tumor heterogeneity on gene expression. However, some issues should be addressed in more details:

- 1) They should perform bulk exome-sequencing also in the other two patient blood samples.
- 2) I find on page 8 an assumption that has no justification, nor statistical or biological ("For the time-series analysis we chose paths in our phylogenetic trees that we perceived to be their backbones."). I think they should at least explain why they have this perception, to give some kind of biological justification if they do not give a statistical one.
- 3) A remark on statistical methodology. I think it is rather "rough" to calculate a correlation based on a division into quartiles (eg. P.12, methodology on page 7-8). I wonder why they did not use a rank correlation on expression values as such, instead of splitting them into four blocks (practically assigning an ordinal value from 1 to 4). Such an approach is out of my knowledge; I'd like to know if it is a well-established methodology (please give references in this case), or if it is the kind of data that is so imprecise as to require such a broad approximation. In any case, the data is transformed semi-quantitatively (measurement of the expression is transformed in an ordinal variable, an integer from 1 to 4), and then the use of the Pearson correlation is not correct, it is necessary to use a nonparametric technique (i.e. Spearman rank correlation). I hope this would not affect the results too much, but for sure it will require checking most of the subsequent analyses.
- 4) At the beginning of the last paragraph of page 14 there is a typo (A "sdeletion" targeting).

Reviewer #3:

Muller and coworkers have undertaken single-cell RNA sequencing of glioblastomas with differential copy-number in PDGF and EGF and are trying to construct phylogenetic maps using RNA-Seq data which they attempt to map back to CNAs by read-count analysis. These are then used to assign subtypes and to try to infer differences in the progression of subtype evolution and changes in specific pathways of aggression. To my knowledge, the work is unique in providing a large single-cell RNA-Seq dataset of several GBMs, and this resource may be the main contribution. The inference of CNAs from a combination of bulk-exome and single-cell RNA-Seq is both challenging and technically dubious, and makes the results very difficult to interpret.

Major

1. Computational Biology - Variant Calling

- * Several pretty standard steps in the exome data-analysis appear to be missing (OxoG contamination, ConTest or equivalent for contamination, comparing genotypes across multiple samples from the same individual, etc.) and should be done & reported.
- * The authors haven't even told us what Nimblegen capture was used, if only on-target reads/SNVs are being-used, and so forth. Significantly better reporting here is required.
- * I could not see where the raw sequencing data for all samples/cells was being made available, but given the critical value of this resource and NPG policy, that needs to be done and an appropriate accession given.
- * There is no demonstration of the accuracy of reported variants, and this is particularly critical for exome-based CNA and indel calls, which are particularly inaccurate
- * It is unclear why the authors excluded cells that were a sub-population lacking a high-frequency CNV. Do they have specific evidence that these are technical artifacts? And, if not, how would they justify exclusion of such a significant amount of data?

2. Writing & Design

The paper is very complex, and was very difficult to follow. After a couple of readings I remain unclear of several key points. First, whether the RNA-Seq data is faithfully representing the copy-number status of cells and how the authors are sure of that. Second, to what extent are phylogenies being created on the basis of expressed point-mutations, and how would this result be reliable? Third, if I understand correctly, the analysis is very indirect: bulk CNAs are called (non-subclonally) from exome data, and then validated using single-cell RNA-Seq. Neither of those platforms are really intended for identifying CNAs, so this contributes both much confusion and much uncertainty about whether the core data is really robust. Consider that the title asks about ordering of mutations and the paper claims to construct phylogenetic maps, but without actually using any genetic information directly.

Minor

- * please give software versions and parameterizations for all algorithms

1st Revision - authors' response

19 July 2016

(Begins on next page)

We would like to thank all the reviewers for their suggestions that led to an improved manuscript. Here is our point-by-point response.

Reviewer #1:

1) Biological validation - the manuscript is entirely bioinformatic. This is not per se a problem, but the paper aims to make functional conclusions, not only correlations, for example, that the new PDGFR mutation is gain of function. Even if a similar deletion in the same domain has been previously shown to promote dimerization, it would greatly strengthen the paper to show that this mutation is gain of function. Similarly, the dose dependence of specific transcriptional programs on the mutations is a fascinating inference. However, to conclude this with confidence, it needs to be directly tested by examining gene expression, or at least the level of a few key transcripts, in response to different levels of the key EGFR and PDGFR mutations.

We have performed additional functional studies to demonstrate that the novel mutation *PDGFRA* Δ^7 represents a gain of function. We expressed one of *PDGFRA* Δ^7 , wild-type *PDGFRA* and a GFP control from lentivirus in two separate patient-derived glioma cell lines. As we describe on page 8, paragraph 1 and in Figure 6 of the revised manuscript, these cell lines do not endogenously express *PDGFRA* above background levels. But, ectopic over-expression of *PDGFRA* Δ^7 provided a growth advantage (as measured via WST-1 colorimetry assay) compared to both wild-type *PDGFRA* over-expression and GFP control. We performed RNA sequencing of these cultures and analysis-of-deviance to identify genes and pathways specifically upregulated by *PDGFRA* Δ^7 . We found an up-regulation of the PDGF and VEGF pathways that matched the correlative, in vivo single-cell analysis we had performed previously. Intriguingly, we also saw an up-regulation of mRNA from the EGFR/Erb-B2 ligand epiregulin, and myeloid cell chemotactic factors *COX2*, *CSF3* and others. To our knowledge, these genes have not been shown previously to be downstream of the wild-type PDGF-receptor. Additionally, we designed a PCR assay with a probe targeted to the deleted region in *PDGFRA* Δ^7 . In both cell lines, ectopic expression of *PDGFRA* Δ^7 induces the expression of wild-type *PDGFRA*. Additionally, we re-analyzed TCGA exome-seq data for GBM. In-frame deletions in the *PDGFRA* I-set domain where *PDGFRA* Δ^7 occurs are frequent events, 18% of cases by our estimates. Taken together, our data indicate that *PDGFRA* Δ^7 enhances PDGF-receptor network gene expression, induces wild-type *PDGFRA* production, and can provide a growth advantage compared to wild-type *PDGFRA*. Considering the population frequency *PDGFRA* in-frame

deletions, these results warrant additional studies to assess the impact of these deletions on receptor tyrosine-kinase inhibitor therapies and their potential as tumor specific targets.

2) Additional relevant literature should be cited, included Francis et al, who show construction of phylogenies from single cell DNA seq in GBM.

We have added the following additional citations in both the introduction (page 3, paragraph 1), and in the discussion (page 11, paragraph 2), providing additional relevant literature regarding phylogenetic reconstruction and other single cell analyses in GBM: Francis *et al*, 2014, Garvin *et al* 2015, Meyer *et al*, 2015, and Navin *et al* 2011.

Reviewer #2:

1) They should perform bulk exome-sequencing also in the other two patient blood samples.

We apologize for the confusion. We had sequenced separate blood controls for each individual patient in the original manuscript, and had used these patient-specific controls to call somatic mutations. We've now clarified this in the revised manuscript (page 3, paragraph 3). "We also performed bulk exome-seq on a separate blood sample from each patient."

2) I find on page 8 an assumption that has no justification, nor statistical or biological ("For the time-series analysis we chose paths in our phylogenetic trees that we perceived to be their backbones."). I think they should at least explain why they have this perception, to give some kind of biological justification if they do not give a statistical one.

Our choice was based on trying to identify an ordering of the dominant clones in the sample. But, in principle, any path in the phylogeny could be analyzed in this way. We chose the path that was comprised by the most frequent mutation at each level of the tree. And, the paths we chose in total comprise 79%, 84% and 44% of cells in samples SF10282, SF10345 and SF10360 respectively. We have revised our description (page 20, paragraph 1) to the following: "For the time-series analysis we chose paths in our phylogenetic trees corresponding to the most frequent mutation at each level. Our goal was to identify an ordering of the dominant clones in the sample. But, in principle expression along an arbitrary path can be measured."

3) A remark on statistical methodology. I think it is rather "rough" to calculate a correlation based on a division into quartiles (eg. P.12, methodology on page 7-8). I wonder why they did not use a rank correlation on expression values as such, instead of splitting them into four blocks (practically assigning an ordinal value from 1 to 4). Such an approach is out of my knowledge; I'd like to know if it is a well-established methodology (please give references in this

case), or if it is the kind of data that is so imprecise as to require such a broad approximation. In any case, the data is transformed semi-quantitatively (measurement of the expression is transformed in a ordinal variable, an integer from 1 to 4), and then the use of the Pearson correlation is not correct, it is necessary to use a nonparametric technique (i.e. Spearman rank correlation). I hope this would not affect the results too much, but for sure it will require checking most of the subsequent analyses.

Although our approach was ad hoc, our rationale was to smooth inter-cellular variation, in order to identify pathways correlating only with *PDGFRA*^{Δ7}. We chose Pearson correlation as opposed to Spearman correlation, since we were interested in identifying linear dependencies on *PDGFRA*^{Δ7} as opposed to strictly monotonic relationships. None the less, we feel that the reviewer makes a valid argument. We have re-done the dose-response and downstream analysis using the reviewer's suggested approach of computing Spearman rank correlation on the expression values. We found that both the pathway enrichment and the transcription factor network analysis produced nearly identical results (see page 7, paragraph 1; page 19, paragraph 1; Figures 4 and 5). In particular, supporting our main conclusion, 100% of the genes enriched for PDGF-receptor pathway in the old analysis are also statistically enriched when we performed the analysis according to the reviewer's recommendation (Fig 4B).

4) At the beginning of the last paragraph of page 14 there is a typo (A "sdeletion" targeting).

We've corrected the error.

Reviewer #3:

Major

1. Computational Biology - Variant Calling

* Several pretty standard steps in the exome data-analysis appear to be missing (OxoG contamination, ConTest or equivalent for contamination, comparing genotypes across multiple samples from the same individual, etc.) and should be done & reported.

We've performed the additional analyses that the reviewer has requested, and reported the results in the revised manuscript (page 15, paragraph 1 and Dataset EV9). In summary, ConTest cross-individual contamination was extremely low: SF10345:0.1%, SF10360:0.1%, SF10282:0.2%, and oxidation error probabilities were exceedingly small: SF10282: 2.8e-5, SF10345: 1.3e-5, SF10360: 6.2e-5. As for comparing genotypes across multiple samples from the same individual, that is not feasible in this study. These biopsies are resected from brain tumors, and represent a limited, precious resource. Although we would like from a scientific

perspective to have multiple samples, from the perspective of patient care the size and number of samples produced by brain surgeries such as these are typically limited.

We used a well-established exome-seq processing pipeline in the original manuscript, *the identical pipeline* was used in several recent, high-impact publications: Mazor et al. *Cancer Cell*. 2015, Johnson et al. *Science*. 2014, van Thuijl et al. *Acta neuropathologica*. 2015.

* The authors haven't even told us what Nimblegen capture was used, if only on-target reads/SNVs are being-used, and so forth. Significantly better reporting here is required.

We've inserted information regarding the kit and the filtering criteria into the manuscript (page 14, paragraph 3).

* I could not see where the raw sequencing data for all samples/cells was being made available, but given the critical value of this resource and NPG policy, that needs to be done and an appropriate accession given.

The RNA-seq and Exome-Seq data from this publication has been deposited at the European Genome-phenome Archive (EGA, <http://www.ebi.ac.uk/ega/>) which is hosted at the EBI, under accession number EGAS00001001900. We have added this information to the manuscript (page 21, paragraph 2).

* There is no demonstration of the accuracy of reported variants, and this is particularly critical for exome-based CNA and indel calls, which are particularly inaccurate

Fresh tissue from brain tumor surgeries is typically limited by the extent of safe resection, as was the case for our study. We do not have the material to perform extensive validation of all variants identified in our exome sequencing. Our level of validation is consistent with other recent studies involving exome-seq of GBM (e.g. Mazor et al. 2015, van Thuijl et al. 2015). That being said, our conclusions center around the *PDGFRA* Δ^7 deletion and the CNVs. *PDGFRA* Δ^7 is indeed validated by the fact that it is identified in two orthogonal platforms: exome-seq and single-cell mRNA-seq (Fig 4A). In both, the deletion is supported by high-quality split alignments, with robust base calls.

To satisfy the reviewer's concerns of CNV accuracy from exome-seq, we ran a CGH array on the tumor and blood control samples of SF10360 (for which we had additional material). Oligo aCGH was carried out using Human Genome microarrays (Agilent), consisting of 180,000 60-mer oligonucleotide DNA probes with approximately 13 kb average spatial resolution. Five

Circos plot of somatic copy number alterations detected in the bulk DNA of SF10360 using exome-seq (outer ring) and aCGH array (inner ring). Regions affected by copy number alterations are highlighted with black dots. Green background indicates chromosomal loss, red background chromosomal gain. For exome-seq, filtering as described in the material/methods section was applied. For aCGH array, we applied a p-val threshold of $p < 1E-30$.

hundred nanograms of tumor and blood reference DNA was used for labeling. The manufacturer's protocol was followed. Microarrays were scanned at 5 μ m resolution using an Agilent-G25005B scanner. All samples passed Agilent quality control assessment. Feature-level data was abstracted with Agilent Feature Extraction software.

An intersection of the CNV calls made by the CGH array and the exome-seq data are in near-perfect agreement. If we calculate the total number of bases detected as altered by copy number by any of the two techniques as 625,940,701 bases, of which 588,494,052 are detected

by both, we come to an agreement of 94.02%. This number becomes more dramatic if we consider all bases that agree in the sense that no CNV is detected for them by both techniques. In this case, the agreement is over 99%.

Moreover, exome-seq for CNV calling is widely used (e.g. Zack et al. Nature Genetics. 2013, Witkiewicz et al. Nature Communications. 2015, Wang et al. Nature Genetics. 2015. Krumm et al. Genome Res. 2012). For example, take Koboldt et al. "VarScan 2: somatic mutation and copy number alteration discovery in cancer by exome sequencing". Genome Res. 2012. Consider Figure 2, panel B from that publication. The authors show that "A total of 206 large-scale CNAs were detected, of which 165

Koboldt et al. Fig 2B. Intersection of large-scale copy number variants detected by SNP array whole-genome sequencing and exome-sequencing for 5 high-grade ovarian tumors.

(80.10%) were detected by all three approaches....". Therefore, we have no doubts, that the copy number alterations we detect in our samples represent real events and not technical artifacts.

* It is unclear why the authors excluded cells that were a sub-population lacking a high-frequency CNV. Do they have specific evidence that these are technical artifacts? And, if not, how would they justify exclusion of such a significant amount of data?

Chromosome 7 gain in SF10345, chromosome 4q12 gain in SF10282 and chromosome 10 loss in SF10360 are nearly ubiquitous and are therefore representative of the dominant clone in their respective biopsies. This is consistent with the fact that they affect cancer driver genes *EGFR*, *PDGFRA* and *PTEN* respectively. Those small fractions of cells lacking the near-ubiquitous mutation of the dominant clone are most likely minor sub-clones from a lineage that is under-sampled in our biopsy. Their common ancestor with the dominant clone of the biopsy (the clone that does have the mutation in chr7, ch4, etc...) cannot be resolved from the information in one biopsy alone. It is also possible that those cells represent technical artifacts, but we have no evidence to that effect (their quality control metrics and images of the capture sites are satisfactory). We've re-written the phylogeny methods section in the revised manuscript (page 17, paragraph 3): "We identified 5-6 cells per sample that did not harbor the CNV with the highest frequency (chromosome 7 gain in SF10345, chromosome 4q12 gain in SF10282 and chromosome 10 loss in SF10360). Since these mutations are nearly ubiquitous, they represent founding mutations for the dominant clones sampled in our cells. This is consistent with the fact that they affect cancer driver genes *EGFR*, *PDGFRA* and *PTEN* respectively. These rare cells lacking founding mutations may be technical outliers, or members of a lineage under-sampled in the biopsy."

2. Writing & Design

The paper is very complex, and was very difficult to follow. After a couple of readings I remain unclear of several key points. First, whether the RNA-Seq data is faithfully representing the copy-number status of cells and how the authors are sure of that.

We are very confident that our triage of single-cell libraries, according to the presence or absence of CNVs called from exome-seq data, is accurate. Firstly, *we threshold in a principled fashion*, by comparing reads mapping a CNV region in a tumor cell to the distribution of reads mapping that CNV across an ensemble of non-malignant human brain cells. This distribution represents a wide variety of cell types found in the brain. As a consequence of our threshold, a

Patel et al. *Science*. 2014. Fig 1C. Heatmap of CNV signal in single-cell RNA-seq data.

randomly chosen non-malignant brain cell would have only a 1/20 chance of being misclassified as possessing a mutation. Secondly, estimating genomic amplification from single-cell transcriptomics data has been done previously with a similar approach, and estimated alterations reflected common high-frequency genomic alterations in

GBM (Patel et al. 2014). Third, wavelet analysis was used to extract the trend line of gene-expression across the chromosome body. We found that these trend lines were systematically elevated or depressed in regions of copy-number alterations, but were stochastic and flat otherwise. This is consistent with numerous studies showing correlation between large scale CNVs and gene expression at the bulk and single-cell level, i.e. Peña-Llopis et al. *Nature Protocols*. 2013. Hou et al. 2016 show that RNA-seq trend lines correlate with CNV status both

Hou et al. *Cell Research*. 2016. Fig 3. Correlations between copy number alterations and gene expression trend lines.

Fig 2B,C from manuscript, showing agreement between single-cell RNA-seq trend lines and exome-seq CNVs

at the single-cell and bulk level (see Fig 3 above, Fig S4 and S5).

Second, to what extent are phylogenies being created on the basis of expressed point-mutations, and how would this result be reliable?

Expressed point mutations are not used to construct phylogenies.

Third, if I understand correctly, the analysis is very indirect: bulk CNAs are called (non-subclonally) from exome data, and then validated using single-cell RNA-Seq. Neither of those platforms are really intended for identifying CNAs, so this contributes both much confusion and much uncertainty about whether the core data is really robust. Consider that the title asks about ordering of mutations and the paper claims to construct phylogenetic maps, but without actually using any genetic information directly.

Exome-seq is widely used to identify copy number alterations (e.g. Zack et al. Nature Genetics. 2013, Witkiewicz et al. Nature Communications. 2015, Wang et al. Nature Genetics. 2015, Krumm et al. Genome Res. 2012). CNV calls from exome-seq have a high degree of overlap with calls based on CGH array and whole-genome sequencing, as evidenced by our CGH validation experiment of SF10360 (summarized in the above mandala) and Koboldt et al.2012. Large scale copy-number alterations are represented in single-cell RNA-seq data as demonstrated by Peña-Llopis et al. 2013, Patel et al. 2014, and Hou et al. 2016 (above), as well as by our own analysis (page 5, paragraph 2, Fig 2B,C). Lastly, we triage cells by CNV presence/absence using a principled control distribution representing a diverse population of non-malignant human cells (Fig 2A).

Minor

* please give software versions and parameterizations for all algorithms

Response:

We attached the algorithm parameters in the Appendix.

Thank you for submitting revised manuscript to Molecular Systems Biology. We have now heard back from the three referees who were asked to evaluate the study. As you will see below, reviewers #1 and #2 think that the study is now suitable for publication. However, reviewer #3 still lists some remaining concerns on the CNA inference based on single-cell RNA-seq data, which we would ask you to address in a revision. In line with the comments of reviewer #3 we would also ask you to make sure that the computational analyses are described in sufficient detail and that the EGA accession number is correct.

REFEREE COMMENTS

Reviewer #1:

The authors have done a commendable job of addressing the critiques. The paper is a valuable contribution to the literature.

Reviewer #2:

The authors have significantly revised their manuscript. The points raised in the previous review have been satisfactorily addressed and the points raised by the other referees have been answered.

Reviewer #3:

The authors have modified the manuscript to address one of my technical concerns (CNA-inference from bulk exomes), but not the other (CNA inference from the single-cell RNA-seq data), and the rebuttal often comprises of "other people have made this work", which is unfortunately not convincing in the case of a cutting-edge technology like single-cell RNA-seq. Further, the concerns about trying to infer phylogeny from RNA data without recourse to the actual underlying genomics of the tumours remain. Further, several of the changes reported as made in the rebuttal letter (e.g. EGA deposition, inclusion of software version numbers, etc.) do not appear to actually have been made.

1. Variant-Calling

- * Exome variant-calling appears to be improved, although the applicants still do not compare the genotypes from the single-cell RNA-seq data to one another or to the exome to demonstrate that they are dealing with cells from the same individual, leaving concerns on data quality
- * There remains no validation of indels in the exome data, outside of the PDGFRA deletion, and global comparisons of DNA- and RNA-sequencing data are now routine in these types of studies at top-tier journals like MSB
- * Incidentally (put likely minor), the comparison of exome- and array-based CNA calls does not appear to require changes in the same direction. Perhaps this is simply unclear writing in the rebuttal letter.
- * While the appendix helps in giving at least some details on algorithms & versions, it is still not reproducible, for example the bwa version is unclear, I think from file-paths to local resources we can infer the GATK version but this isn't clear either, and so forth. Much clearer description needs to be given.
- * Further challenging data-quality, there remains a significant sub-population of cells that lack high-frequency SNVs. The authors suggest that their pipeline is robust and error-free, but then suggest that these may be technical artifacts. Given that exome-sequencing can infer subclonality of CNAs, it is surprising that this hasn't been done here. Further, the authors root their tree in a normal, but actually have the CNA profile of the normal (including germline CNAs) from their exome sequencing so could root the tree properly if they wish. Finally, just because mutations are highly prevalent ("near ubiquitous" in the terminology of the paper), that doesn't make them necessarily "founding mutations for the dominant clones sampled in our cells". To the contrary, high-prevalence

isn't the same thing as a founding mutation, and that would only be the case if these were truly present in 100% of all cells. The study comprises ~300 single cells, and 15-20 show this artifact, suggesting up to 5-6% of cells contain either an artifact or some biology being ignored here.
* The reported EGA accession is not found either by google or by searching at EGA itself.

2. RNA-Seq and CNA Status

The applicants suggest that they are "very confident that our triage of single-cell libraries... is accurate". They provide three arguments why this is the case. First, they suggest that there is a 5% chance of error due to statistical concerns (similar in magnitude to the apparent 5% of cells containing an artifact, providing a net error of 10%). Second, they argue that others have successfully inferred CNAs from RNA-seq, but unfortunately for a technique like single-cell RNA-seq there are significant differences between a paper published two years ago and this one, and simply saying "others have seen this correlation in a few samples" does not provide strong evidence that it holds here. Third, they suggest that a wavelet-based segmentation looks appropriate for a CNA. This still doesn't provide strong evidence that this calling is correct. This is the absolute heart of the paper: the authors want to infer phylogeny from RNA data rather than DNA data, and thus the correct inference of CNAs from RNA-seq is fundamental.

Minor

* Figure 2C uses red and green together, which is almost unreadable for colour-blind people and NPG recommends against this colour combination

2nd Revision - authors' response

16 August 2016

(Begins on next page)

We are very grateful to all 3 reviewers, whose thoughtful comments have led to a much improved manuscript. We are very happy to hear that Reviewers #1 and #2 feel the paper is ready for publication. Reviewer #1's comment, "*The authors have done a commendable job of addressing the critiques. The paper is a valuable contribution to the literature.*", is particularly gratifying. We were also glad to see that Reviewer #2 feels: "*The authors have significantly revised their manuscript. The points raised in the previous review have been satisfactorily addressed and the points raised by the other referees have been answered.*". We have performed additional analysis to address the concerns of Reviewer #3, which we describe below in a point-by-point response.

Reviewer #3:

The authors have modified the manuscript to address one of my technical concerns (CNA-inference from bulk exomes), but not the other (CNA inference from the single-cell RNA-seq data), and the rebuttal often comprises of "other people have made this work", which is unfortunately not convincing in the case of a cutting-edge technology like single-cell RNA-seq.

We are gratified to hear that we have addressed Reviewer 3's technical concerns, regarding CNA inference from exome-seq. We want to underscore that we do not infer CNA from single-cell RNA-seq data. Rather, we classify single-cell RNA-seq libraries according to the presence or absence of CNAs inferred from exome-seq. In this revision, we have we have included new analyses to address this comment. Please see our response to section 2 below.

Further, the concerns about trying to infer phylogeny from RNA data without recourse to the actual underlying genomics of the tumours remain.

Genomic data (exome-seq) is critically important for our approach. The copy-number calls made from exome-seq are the basis of our phylogenies. We feel the strength of our approach is its integration of genomic and transcriptomic data. We have performed additional analyses to address Reviewer 3's particular concerns.

Further, several of the changes reported as made in the rebuttal letter (e.g. EGA deposition, inclusion of software version numbers, etc.) do not appear to actually have been made.

We apologize for any misunderstanding. We had deposited the data with EGA (EGAS00001001900), but EGA has not yet made the study publicly accessible. As described below, we are making inquiries with EGA to expedite its release and we are happy to provide the raw data to the Reviewers via the editor.

We also apologize if any of our code was unclear or inadequately commented. We've revised the Appendix for clarity.

* Exome variant-calling appears to be improved, although the applicants still do not compare the genotypes from the single-cell RNA-seq data to one another or to the exome to demonstrate that they are dealing with cells from the same individual, leaving concerns on data quality

We have addressed this concern in the following manner: To compare genotypes from the single-cell RNA-seq data across patients and to the exome-seq data, we identified 17, 21 and 15 single-nucleotide variants (SNVs) in the single-cell RNA-seq that are patient specific in SF10282, SF10345 and SF10360 respectively. These SNVs are likewise detected in, and only in, their respective blood and tumor-derived exome-seq datasets. The median difference in RNA-seq to tumor exome-seq variant allele frequency (VAF) is 0.056. By comparison, the median difference in tumor exome-seq to blood exome-seq VAF is 0.044 (see Appendix S3). These data, and the minimal contamination observed in our revision #1 response, are consistent with the fact that the tumor samples from each patient were processed on separate

days. Fresh tumor tissue was minced, an aliquot was removed and a DNA extraction performed for exome-seq. An aliquot of the same mince was disassociated into single-cell suspension and immediately loaded into a Fluidigm C1 capture chip. We have added a discussion of this genotype quality check to the Materials and Methods section, page 16 paragraph 2, and we have added an associated figure to the Appendix (S3 and S4).

* There remains no validation of indels in the exome data, outside of the PDGFRA deletion, and global comparisons of DNA- and RNA-sequencing data are now routine in these types of studies at top-tier journals like MSB

We used Pindel to screen for potential indels and chose to focus on *PDGFRA* Δ^7 . It was not our intention to perform a comprehensive study of indels in these data, but we reported all of the Pindel calls in the appendix for completeness. In our revised manuscript we have annotated each of the Pindel calls in Datasets EV 1-3 to denote whether the call is not supported by the RNA-seq, is not supported by the RNA-seq data or if there is no coverage in the RNA-seq data. This additional annotation should address the Reviewer's concern regarding DNA-RNA indel comparisons.

* Incidentally (put likely minor), the comparison of exome- and array-based CNA calls does not appear to require changes in the same direction. Perhaps this is simply unclear writing in the rebuttal letter.

Yes, we required CNA calls to be in the same direction when comparing exome-seq and array-based calls in the previous rebuttal.

* While the appendix helps in giving at least some details on algorithms & versions, it is still not reproducible, for example the bwa version is unclear, I think from file-paths to local resources we can infer the GATK version but this isn't clear either, and so forth. Much clearer description needs to be given.

We apologize for any lack of clarity in the appendix, we have re-written this section.

* Further challenging data-quality, there remains a significant sub-population of cells that lack high-frequency SNVs. The authors suggest that their pipeline is robust and error-free, but then suggest that these may be technical artifacts.

It is possible that those cells represent technical artifacts. We have no evidence to that effect. The quality control metrics for these cells and the images of their capture sites are satisfactory. We believe that those cells are from a lineage that was under-sampled in our biopsy.

Given that exome-sequencing can infer subclonality of CNAs, it is surprising that this hasn't been done here.

We ran Sequenza (Favero et al. 2015) to estimate the VAFs of the founding CNAs of the dominant clones: chromosome 7 gain in SF10345 (7+), chromosome 4q12 gain in SF10282 (4q12+) and chromosome 10 loss in SF10360 (10-). The expected VAFs are 94% for 4q12+, 85% for 7+ and 95% for 10-. These estimates indicate these mutations are clonal, which is consistent with their prevalence in the single-cell RNA-seq data. This additional analysis should address the Reviewer's concern about the clonality of the mutations that we use to define the dominant clones.

Further, the authors root their tree in a normal, but actually have the CNA profile of the normal (including germline CNAs) from their exome sequencing so could root the tree properly if they wish.

We were interested in somatic mutations accumulated during cancer progression and not germline mutations. So, we did not annotate germline CNAs in our phylogenies (which would be present in every branch of the tree). We ran each blood sample through our CNV detection pipeline. We did not detect any germline copy-number alterations in any of our samples. This additional analysis should address the Reviewer's concern about germline CNAs.

Finally, just because mutations are highly prevalent ("near ubiquitous" in the terminology of the paper), that doesn't make them necessarily "founding mutations for the dominant clones sampled in our cells". To the contrary, high-prevalence isn't the same thing as a founding mutation, and that would only be the case if these were truly present in 100% of all cells. The study comprises ~300 single cells, and 15-20 show this artifact, suggesting up to 5-6% of cells contain either an artifact or some biology being ignored here.

Chromosome 7+ in SF10345, 4q12+ in SF10282 and 10- in SF10360 are present in over 90% of cells in the single-cell RNA-seq data. Taken together with the above Sequenza analysis, this indicates that they are representative of the dominant clone their respective biopsies. They are founding mutations of the dominant clones as we have defined them, since they are in 100% of those cells. This is consistent with the fact that they effect important oncogenes/tumor suppressor *EGFR*, *PDGFRA* and *PTEN* respectively. We believe that those small fractions of cells lacking the founding mutation of the dominant clone are sub-clones from a lineage that is under-sampled in our biopsy. The common ancestor of that lineage with the dominant clone cannot be resolved from the information in one biopsy alone.

* The reported EGA accession is not found either by google or by searching at EGA itself.

The data has been deposited with EGA, under the accession EGAS00001001900. The study has not yet been publicly released by EGA. We apologize for the delay and for any misunderstanding. We have requested a status update from EGA and will notify the Editor as soon as the study is released. We are happy to provide any additional information to assure the Editor and Reviewers that the data are deposited and will be released. And, of course the Reviewers are welcome to the raw data at any time during the review process through a personal communication via the Editor.

2. RNA-Seq and CNA Status

The applicants suggest that they are "very confident that our triage of single-cell libraries... is accurate". They provide three arguments why this is the case. First, they suggest that there is a 5% chance of error due to statistical concerns (similar in magnitude to the apparent 5% of cells containing an artifact, providing a net error of 10%). Second, they argue that others have successfully inferred CNAs from RNA-seq, but unfortunately for a technique like single-cell RNA-seq there are significant differences between a paper published two years ago and this one, and simply saying "others have seen this correlation in a few samples" does not provide strong evidence that it holds here. Third, they suggest that a wavelet-based segmentation looks appropriate for a CNA. This still doesn't provide strong evidence that this calling is correct. This is the absolute heart of the paper: the authors want to infer phylogeny from RNA data rather than DNA data, and thus the correct inference of CNAs from RNA-seq is fundamental.

We feel we've controlled the error rate in a principled fashion. We've performed additional analysis to demonstrate the low false-discovery rate of our classifier. Additionally, we demonstrate high correlations between exome-seq and single-cell RNA-seq read densities. The fact that 3 labs, Patel et al. 2014, Hou et al. 2016 and now ours, have independently observed the same phenomenon (despite differences in cell capture approach or cancer types) supports the notion that the correlation between CNA and gene expression is real.

To provide three new points of evidence that our method is robust, we have done the following:

1. Firstly, we computed read densities in the single-cell RNA-seq samples and in the corresponding tumor exome-seq sample. We find that the mean Pearson correlations are 0.896, 0.739 and 0.742 for SF10282, SF10345 and SF10360 respectively (see Appendix S1). Similar results were found in Patel et al. 2014 and Hou et al. 2016.
2. We have computed the false discovery rate of our method via 10-fold cross validation and we find the median false discovery rates to be 0.4%, 0.46% and 0.35% for SF10282, SF10345 and SF10360 respectively (see Appendix S2A).
3. Lastly, we have computed false discovery rates by applying our triage algorithm to non-malignant fetal brain cells which were single-cell RNA-sequenced and recently published in Pollen et al. 2016. We found the median empirical false discovery rates to be 4.1%, 4.2% and 2.7% for SF10282, SF10345 and SF10360 respectively (see Appendix S2B).

This new analysis, together with our original 3 points of evidence, we feel provides strong support for the validity of our method, and should address the Reviewer's concerns.

Minor

* Figure 2C uses red and green together, which is almost unreadable for colour-blind people and NPG recommends against this colour combination

We apologize for our poor choice of color combination. We have changed the figure's colors to magenta and green to accommodate the color blind, as recommended in Wong. *Nature Methods* 2011.

Thank you again for sending us your revised manuscript. We have now heard back from reviewer #3 who was asked to evaluate the revised study. As you will see below, the reviewer still raises significant concerns on your work, which unfortunately preclude its publication in *Molecular Systems Biology*.

Reviewer #3 is still not convinced that his/her concerns regarding the genomic data analysis and CNV inference have been satisfactorily addressed. As such, s/he indicated that s/he does not support publication of the study in *Molecular Systems Biology*.

Please allow me to mention that we have additionally consulted a member of our Editorial Advisory Board, who is an expert on genomic data analysis. S/he agreed with the concerns of Reviewer #3 and mentioned that it is indeed likely that there is some bias in the CNV calling and that additional analyses/validation is required to support the related conclusions.

Considering the significant concerns that remain and the fact that substantial additional analyses with unclear outcome are required to support key conclusions of the study, I see no other choice than to return the manuscript with the message that we cannot offer to publish it. In any case, thank you for the opportunity to examine your work. I hope that the points raised will prove useful to you and that you will not be discouraged from submitting future work to *Molecular Systems Biology*.

REFEREE REPORTS

Reviewer #3:

Unfortunately the authors have generally not addressed head-on most of the concerns with this manuscript regarding validation of their results. Their CNA calls are uncertain from the exome data (in this revision the authors reveal they do not find a single germline CNV across three samples, which is extremely unlikely given the population distribution), and from the RNA-seq data (where no direct validation is given). These issues are at the core of the reliability of the manuscript. The authors must provide clear validation of their exome CNV and RNA-seq CNV classifications in their own samples using independent techniques, not recourse to literature studies.

"We are gratified to hear that we have addressed Reviewer 3's technical concerns, regarding CNA inference from exome-seq. We want to underscore that we do not infer CNA from single-cell RNA-seq data. Rather, we classify single-cell RNA-seq libraries according to the presence or absence of CNAs inferred from exome-seq. In this revision, we have we have included new analyses to address this comment. Please see our response to section 2 below."

The authors appear to be making a distinction that doesn't have meaning. To classify patients based on presence or absence of a CNA is functionally interchangeable with inferring this CNA. Thus the core concerns remain unchanged.

"Genomic data (exome-seq) is critically important for our approach. The copy-number calls made from exome-seq are the basis of our phylogenies. We feel the strength of our approach is its integration of genomic and transcriptomic data. We have performed additional analyses to address Reviewer 3's particular concerns."

Again, this not really evidence. It is an opinion that by starting off with genomic data (CNAs from a single bulk exome sample from different cells than those on which RNA-seq was performed) can provide a basis for phylogenetic reconstruction based on transcriptomic data. The core concern remains.

"It is possible that those cells represent technical artifacts. We have no evidence to that effect. The quality control metrics for these cells and the images of their capture sites are satisfactory. We believe that those cells are from a lineage that was under-sampled in our biopsy."

"We ran Sequenza (Favero et al. 2015) to estimate the VAFs of the founding CNAs of the dominant clones: chromosome 7 gain in SF10345 (7+), chromosome 4q12 gain in SF10282 (4q12+) and chromosome 10 loss in SF10360 (10-). The expected VAFs are 94% for 4q12+, 85% for 7+ and 95% for 10-. These estimates indicate these mutations are clonal, which is consistent with their prevalence in the single-cell RNA-seq data. This additional analysis should address the Reviewer's concern about the clonality of the mutations that we use to define the dominant clones."

This remains one of the core problems that seriously jeopardizes the correctness of the paper. The authors continue to suggest that their overall pipeline is robust, but multiple sources of error are accumulating. There is this population of cells lacking high-frequency SNVs. Then the exome CNA analysis suggests that up to 15% of cells do not harbour the mutations. And then their algorithms to classify have a 5% per cell (no multiple testing adjustment for that step!) error rate. These accumulating errors suggest an overall CNA misclassification rate that may approach 25% or more. There is still no robust consideration of error rates and no quantification of how it will influence the final results or conclusions of this paper. Without this error in cell classification issue being addressed, it will not be possible to evaluate the correctness of the results presented. The authors have to give some robust validation of the accuracy of this prediction, which is central to their paper.

"We were interested in somatic mutations accumulated during cancer progression and not germ-line mutations. So, we did not annotate germ-line CNAs in our phylogenies (which would be present in every branch of the tree). We ran each blood sample through our CNV detection pipeline. We did not detect any germ-line copy-number alterations in any of our samples. This additional analysis should address the Reviewer's concern about germ-line CNAs."

Unfortunately now this additional analysis raises critical concerns around the quality of the exome CNV calling. The 1kg project identified a median of ~20 Mbp of variation per individual in its low-resolution data, after covering ~75% of the genome (<http://www.nature.com/nature/journal/v526/n7571/full/nature15394.html>). The exome capture kit here covers 64 Mbp of DNA. This implies that the authors should be seeing in their 64 Mbp capture kit: $64 / (3234 \times 0.75) \times 20 = 0.52$ Mbp of structural variation at a median per sample. That works out to approximately 150 genes/sample. This is of course only an estimate, but to see zero in all three samples strongly suggests serious problems with the exome CNV calling pipeline, as initially suggested, and highlights the urgent need for proper validation.

"Firstly, we computed read densities in the single-cell RNA-seq samples and in the corresponding tumor exome-seq sample. We find that the mean Pearson correlations are 0.896, 0.739 and 0.742 for SF10282, SF10345 and SF10360 respectively (see Appendix S1). Similar results were found in Patel et al. 2014 and Hou et al. 2016. "

As a further highlight, a correlation of 0.75 is actually quite poor, since the variance explained will be the square of the correlation, which here is only 0.5625. Of course there is some non-zero correlation between RNA-seq and exome CNAs, but the question is the ability to predict those at high accuracy, and with ~40-45% of variance unexplained that remains in question.

RE: MSB-16-6969RRR

Dear Reviewers and Editors:

We would like to thank the reviewers and editors for the time they have committed to the review process. To be candid, we were all quite surprised to read the negative opinion of reviewer #3. We believe that this negative opinion has arisen from two issues: (1) the reviewer's opinion about the reliability of CNV calls has "reversed" during the course of review, and (2) the size of CNVs used in our study are much larger than those reviewer #3's objections are based on.

With regard to the first point: In their initial review, reviewer #3 suggested a genomic hybridization array as a gold standard to evaluate CNV accuracy. We therefore performed this validation, and when the array showed excellent agreement, reviewer #3 confirmed: "The authors have modified the manuscript to address one of my technical concerns (CNA inference from bulk exomes) ...". But, in the latest review, our CNV inference pipeline is again called into question anew. This "reversal" of opinion about the reliability of CNV calls has made their question a bit of a moving target.

About the second point: Our CNV pipeline is designed to discover mega-base scale CNVs (median 18-21 mega base-pairs, comprising 300-400 genes), which are not common in the germline. Reviewer #3 perhaps believes that we are studying small amplifications/deletions, which might be expected in the germline. In the reference the reviewer provided, the median alteration size was 36 kilo base-pairs, comprising only 1 gene. Our analyses only utilize large CNVs that we can reliably detect (as described in Methods).

Lastly, reviewer #3 states "There is still no robust consideration of error rates...". But, in our last revision we performed cross-validation, as well as empirical testing on an external dataset, to estimate false-discovery rates. The estimated false-discovery rate is <0.06 in all tests. Reviewer #3 requested this analysis, but has not commented on it (Appendix 2).

We appreciate the opportunity to address reviewer #3's comments point-by-point, which we do below:

Reviewer #3:

Unfortunately the authors have generally not addressed head-on most of the concerns with this manuscript regarding validation of their results. Their CNA calls are uncertain from the exome data (in this revision the authors reveal they do not find a single germline CNV across three samples, which is extremely unlikely given the population distribution), and from the RNA-seq data (where no direct validation is given). These issues are at the core of the reliability of the manuscript. The authors must provide clear validation of their exome CNV and RNA-seq CNV classifications in their own samples using independent techniques, not recourse to literature studies.

Reviewer #3 has "reversed" their opinion of our ability to infer CNVs from exome-seq. This is perhaps due their assumption about the size of the CNVs being studied in our manuscript. In the reference provided by reviewer #3, the median alteration size is only 36 kilo base-pairs (affecting 1 gene). In our study, we use large, mega-base scale CNVs that we can reliably detect, median size 18-21 mega base-pairs (comprising 300-400 genes). Alterations of this size are not common in the germline. Lastly, reviewer #3 says that no direct validation is given of our

single-cell classifier. But, in our last revision we performed cross-validation and empirical testing on external datasets to estimate error rates (Appendix 2). Reviewer #3 has not commented on this analysis, which we repeat below.

"We are gratified to hear that we have addressed Reviewer 3's technical concerns, regarding CNA inference from exome-seq. We want to underscore that we do not infer CNA from single-cell RNA-seq data. Rather, we classify single-cell RNA-seq libraries according to the presence or absence of CNAs inferred from exome-seq. In this revision, we have we have included new analyses to address this comment. Please see our response to section 2 below."

The authors appear to be making a distinction that doesn't have meaning. To classify patients based on presence or absence of a CNA is functionally interchangeable with inferring this CNA. Thus the core concerns remain unchanged.

The difference, which we want to highlight, is that we don't identify CNVs from RNA-seq data de novo. Mega-base scale CNVs are determined from exome-seq. We then classify single cells according to the presence or absence of these CNVs, using a classifier with a known error-rate (see below).

"Genomic data (exome-seq) is critically important for our approach. The copy-number calls made from exome-seq are the basis of our phylogenies. We feel the strength of our approach is its integration of genomic and transcriptomic data. We have performed additional analyses to address Reviewer 3's particular concerns."

Again, this not really evidence. It is an opinion that by starting off with genomic data (CNAs from a single bulk exome sample from different cells than those on which RNA-seq was performed) can provide a basis for phylogenetic reconstruction based on transcriptomic data. The core concern remains.

The quote above is an excerpt from our response to the reviewer's comment that we "...infer phylogeny from RNA data without recourse to the actual underlying genomics...". We want to emphasize that we do make recourse to the underlying genomics, via our use of exome-seq.

"It is possible that those cells represent technical artifacts. We have no evidence to that effect. The quality control metrics for these cells and the images of their capture sites are satisfactory. We believe that those cells are from a lineage that was under-sampled in our biopsy."

"We ran Sequenza (Favero et al. 2015) to estimate the VAFs of the founding CNAs of the dominant clones: chromosome 7 gain in SF10345 (7+), chromosome 4q12 gain in SF10282 (4q12+) and chromosome 10 loss in SF10360 (10-). The expected VAFs are 94% for 4q12+, 85% for 7+ and 95% for 10-. These estimates indicate these mutations are clonal, which is consistent with their prevalence in the single-cell RNA-seq data. This additional analysis should address the Reviewer's concern about the clonality of the mutations that we use to define the dominant clones."

This remains one of the core problems that seriously jeopardizes the correctness of the paper. The authors continue to suggest that their overall pipeline is robust, but multiple sources of error are accumulating. There is this population of cells lacking high-frequency SNVs. Then the

exome CNA analysis suggests that up to 15% of cells do not harbour the mutations. And then their algorithms to classify have a 5% per cell (no multiple testing adjustment for that step!) error rate. These accumulating errors suggest an overall CNA misclassification rate that may approach 25% or more. There is still no robust consideration of error rates and no quantification of how it will influence the final results or conclusions of this paper. Without this error in cell classification issue being addressed, it will not be possible to evaluate the correctness of the results presented. The authors have to give some robust validation of the accuracy of this prediction, which is central to their paper.

Sequencing cells without mutations is not intrinsically an error. We control for stromal contamination in exome-seq (page 14), and in single-cell sequencing (page 16). Reviewer #3 states that we have not considered the error rate of our cell classifier. However, in our last revision we tested our classifier on external datasets and additionally performed cross-validation, to address the reviewer's concerns about false-discovery rates (Appendix 2). Reviewer #3 has not commented on this analysis. This analysis demonstrates in two independent tests that we control the error rate in a principled way.

Reviewer #3 also states that we do not control for multiple hypothesis testing at this step, but we do (page 17, paragraph 2 "...[we] control for multiple-hypothesis testing using Benjamini-Hochberg correction."). We repeat our analysis here for your convenience. In summary, the empirical FDR is <0.06 across all samples, and the median FDR is 0.04. In the 10-fold cross-validation assessment, the FDR was <0.01.

2. False discovery rate estimation

Figure S2: Distributions of the false discovery rates of CNV calls on single cell RNA-seq data. A) Cross validation on the dataset used as a normal brain control in the manuscript. We randomly selected 10% of the cells as test, and the remaining 90% as training set. For each sample in the test set we generated CNV calls on each of the patient's altered regions defined by exome-seq. The false positive rate was calculated as: #positive CNV calls/#total CNV calls for each of the ten performed classification rounds. B) For each of the non-malignant cells from the dataset of Pollen et al. (2016) we generated CNV calls on each of the patient's altered regions defined by exome-seq. Error rates were calculated as described in A). For each set sample in Pollen et al., false positive rates were calculated independently.

"We were interested in somatic mutations accumulated during cancer progression and not germ-line mutations. So, we did not annotate germ-line CNAs in our phylogenies (which would be present in every branch of the tree). We ran each blood sample through our CNV detection pipeline. We did not detect any germ-line copy-number alterations in any of our samples. This additional analysis should address the Reviewer's concern about germ-line CNAs."

Unfortunately now this additional analysis raises critical concerns around the quality of the exome CNV calling. The 1kg project identified a median of ~20 Mbp of variation per individual in its low-resolution data, after covering ~75% of the genome (<http://www.nature.com/nature/journal/v526/n7571/full/nature15394.html>). The exome capture kit here covers 64 Mbp of DNA. This implies that the authors should be seeing in their 64 Mbp capture kit: $64 / (3234 \times 0.75) \times 20 = 0.52$ Mbp of structural variation at a median per sample. That works out to approximately 150 genes/sample. This is of course only an estimate, but to see zero in all three samples strongly suggests serious problems with the exome CNV calling pipeline, as initially suggested, and highlights the urgent need for proper validation.

We believe that reviewer #3's criticisms are based on a misunderstanding about the size of CNVs under study in our manuscript. We chose to focus exclusively on large, mega-base scale CNVs that we can detect reliably (median 18-21 mega base pairs, comprising 300-400 genes). CNVs of this size are not common in the germline. The genomic alterations described in reviewer #3's reference are much smaller: median 36 kilo base-pairs, comprising 1 gene.

In their first review, reviewer #3 suggested a genomic-hybridization array as a gold standard for CNV detection. We performed this validation and reviewer #3 seemed satisfied with our analysis, and said: "The authors have modified the manuscript to address one of my technical concerns (CNA inference from bulk exomes),...". However, here reviewer #3 seems to "reverse" their opinion of the accuracy of CNV detection from exome-seq. We repeat a summary of our array validation here: We found a 94% overlap in bases altered, as assessed by either technique. If we additionally consider bases where no CNV is detected by both techniques, then the agreement is over 99%.

Circos plot of somatic copy number alterations detected in the bulk DNA of SF10360 using exome-seq (outer ring) and aCGH array (inner ring). Regions affected by copy number alterations are highlighted with black dots. Green background indicates chromosomal loss, red background chromosomal gain.

"Firstly, we computed read densities in the single-cell RNA-seq samples and in the corresponding tumor exome-seq sample. We find that the mean Pearson correlations are 0.896, 0.739 and 0.742 for SF10282, SF10345 and SF10360 respectively (see Appendix S1). Similar results were found in Patel et al. 2014 and Hou et al. 2016. "

As a further highlight, a correlation of 0.75 is actually quite poor, since the variance explained will be the square of the correlation, which here is only 0.5625. Of course there is some non-zero correlation between RNA-seq and exome CNAs, but the question is the ability to predict those at high accuracy, and with ~40-45% of variance unexplained that remains in question.

The median variance explained is indeed 54.8%, comparing patient-matched tumor exome-seq and tumor single-cell RNA-seq. By contrast, the median variance explained with cells from non-malignant brain (Pollen et al. 2014) is only 5.8%. We are confident that we can robustly detect the order-of-magnitude increase in variance explained (from 5.8% to 54.8%), between cells that do and do not have the mega-base scale CNVs used in this study. We feel the cross-validation and empirically measured error rates in Appendix 2 demonstrate that the error in this triage is controlled in a principled way.

We have consulted two Editorial Advisory Board members and asked them to evaluate the manuscript and your point-by-point response to the remaining concerns of reviewer #3. We have also circulated your response to reviewer #3, in case s/he would have further comments and we informed him/her that we would consult with the EAB.

Both EAB members think that the explanations/responses provided in your point by point response are justified and that the study is suitable for publication in *Molecular Systems Biology* (pending minor text modifications, as you will see below). In particular, regarding the concerns on the CNVs EAB member #2 mentioned: "The authors employ an approach to identify mega-base scale CNVs, which Sudmant et al obviously did not do. The focus of Sudmant et al was on much smaller CNV events (typically <50kb) i.e. such occurring in the germline, and I agree with the authors that these would be unlikely to be detectable in exome sequencing data (in contrast to the large somatic CNVs that the authors report in their manuscript). From the data provided by the authors (Figure 2) I actually got the impression that their bulk exome sequencing based approach works reasonably well for the purpose it is used in the manuscript. Further to this, using single cell RNA data to "classify" single cells according to CNV presence/absence, as used by the authors, seems like a reasonable (and quite elegant) approach to me."

EAB member #1, who also agreed that your responses resolve the remaining concerns, suggested the following text modifications, which we would ask you to include in a minor revision of the manuscript:

- The size difference of the analyzed CNVs compared to those normally detected in the germline (Sudmant et al), should be mentioned in the text.
- The false discovery rate estimation should be described in better detail in the main text.

Thank you for your continued interest in our manuscript: MSB-16-6969RRR-Q "Single-cell sequencing maps gene expression to mutational phylogenies in PDGF and EGF driven gliomas". We appreciate the time and effort that you, the reviewers, and the EAB members have put into the review process. We are glad to hear that two EAB members feel the study is suitable for publication in *Molecular Systems Biology*, pending minor text modifications. We have made the requested modifications to the manuscript, which we detail below.

To address EAB member #1's point that we should explicitly mention the size difference between CNVs used our study and those detected in Sudmant et al, and to cite Appendix Figure S1, we have added the following text:

Page 5, paragraph 2: "We chose to focus on large, somatic CNVs of 100 exons or more (Materials and Methods). The median size of CNVs exceeding this threshold was 18-21 mega base-pairs, comprising 300-400 genes. This size is much larger than the size of CNVs previously observed to occur frequently in the germline (Sudmant et al, 2015), which had a median size of 36 kilo base-pairs. We found that GBM to normal-brain control single-cell expression ratios correlated with CNV status (Appendix Fig S1), motivating us to quantify these CNVs in individual cells (Materials and Methods)."

To address EAB member #1's point that we should describe the false discovery rate estimation in the text in better detail, and to cite Appendix Figure S2, we have added the following text:

Page 5, paragraph 2: "We validated the error-rate of this classifier using 10-fold cross-validation, as well as empirical testing on a control dataset (Pollen et al, 2015a), (Appendix Fig S2, Materials and Methods)."

Page 18, paragraph 1: "To estimate the false discovery rate (FDR) of this classification procedure, we performed 10-fold cross-validation using the normal-brain control cells. For each patient, we randomly selected tranches of 10% as test and 90% as training data. We estimated the FDR as (# positive CNV calls)/(# total CNV calls), for each of the 10 folds. We found the FDR to be < 0.01 for all tests (Appendix Fig S2A). As a second estimator of the FDR, we classified the presence of CNVs

on a dataset comprised of nonmalignant, fetal-brain cells (Pollen et al, 2015b), and estimated the FDR as above. We found these FDR estimates to all be < 0.06 (Appendix Fig S2B).”

Lastly, we have corrected our labeling and citation of Appendix Table S1. We have added an acknowledgement to Pamela Paris (UCSF) for consultation on arrays.

Let me again express my gratitude for the time and care you and the Editorial Board members have taken, in the evaluation of our manuscript.

5th Editorial Decision

08 November 2016

Thank you again for sending us your revised manuscript. We are now satisfied with the modifications made and I am pleased to inform you that your paper has been accepted for publication.

Corresponding Author Name: Aaron Diaz
 Journal Submitted to: Molecular Systems Biology
 Manuscript Number: MSB-16-6969